# Explainable Artificial Intelligence (XAI) and Supervised Machine Learning-based Algorithms for Prediction of Surface Roughness of Additively Manufactured Polylactic Acid (PLA) Specimens

Akshansh Mishra [1,*], Vijaykumar S. Jatti [2], Eyob Messele Sefene [3] and Shivangi Paliwal [4]

[1] School of Industrial and Information Engineering, Politecnico Di Milano, 20121 Milan, Italy
[2] Department of Mechanical Engineering, Symbiosis Institute of Technology, Pune 412115, India; vijaykumar.jatti@sitpune.edu.in
[3] Department of Mechanical Engineering, National Taiwan University of Science and Technology, Taipei 10607, Taiwan; d10903802@mail.ntust.edu.tw
[4] Department of Mechanical Engineering, University of Kentucky, Lexington, KY 40506, USA; shivangi.paliwal@uky.edu
* Correspondence: akshansh.mishra@mail.polimi.it; Tel.: +3935-1576-6436

**Abstract:** Structural integrity is a crucial aspect of engineering components, particularly in the field of additive manufacturing (AM). Surface roughness is a vital parameter that significantly influences the structural integrity of additively manufactured parts. This research work focuses on the prediction of the surface roughness of additive-manufactured polylactic acid (PLA) specimens using eight different supervised machine learning regression-based algorithms. For the first time, explainable AI techniques are employed to enhance the interpretability of the machine learning models. The nine algorithms used in this study are Support Vector Regression, Random Forest, XGBoost, AdaBoost, CatBoost, Decision Tree, the Extra Tree Regressor, the Explainable Boosting Model (EBM), and the Gradient Boosting Regressor. This study analyzes the performance of these algorithms to predict the surface roughness of PLA specimens, while also investigating the impacts of individual input parameters through explainable AI methods. The experimental results indicate that the XGBoost algorithm outperforms the other algorithms with the highest coefficient of determination value of 0.9634. This value demonstrates that the XGBoost algorithm provides the most accurate predictions for surface roughness compared with other algorithms. This study also provides a comparative analysis of the performance of all the algorithms used in this study, along with insights derived from explainable AI techniques.

**Keywords:** additive manufacturing; explainable artificial intelligence; machine learning; supervised learning; surface roughness; structural integrity

## 1. Introduction

Machine learning is a subset of artificial intelligence that involves the development of algorithms that enable computer systems to learn from data and improve their performance over time. In other words, machine learning algorithms are designed to identify patterns in data and use these patterns to make predictions or decisions. Machine learning can be classified into three main categories, i.e., supervised learning, unsupervised learning, and reinforcement learning [1–4]. Supervised learning involves the use of labeled data to train machine learning models. In this type of learning, the machine learning algorithm is provided with input and output data pairs and uses this information to learn how to make accurate predictions on new data [5–7]. Unsupervised learning involves the use of unlabeled data to train machine learning models. In this type of learning, the machine learning algorithm is provided with input data only and must identify patterns or

relationships in the data without any guidance [8–10]. Reinforcement learning involves the use of a reward system to train machine learning models. In this type of learning, the machine learning algorithm is rewarded for making correct predictions or taking the correct action, which encourages the algorithm to improve its performance over time [11–13].

Additive manufacturing (AM) is a rapidly growing field that involves the production of complex parts and components using 3D printing technology [14–17]. Machine learning (ML) has emerged as a valuable tool in AM for optimizing various aspects of the process, including design, fabrication, and post-processing. Machine learning algorithms can be used to optimize the designs of parts and components in additive manufacturing. ML algorithms can analyze large amounts of data and identify patterns that can be used to create more efficient designs that reduce material usage, improve performance, and reduce production costs. This information can be used to optimize the process parameters and improve the quality and consistency of the printed parts [18–20]. Machine learning algorithms can also be used for quality control in additive manufacturing. ML algorithms can analyze images of printed parts and identify defects or anomalies that may affect their performance.

Surface roughness is an important parameter that affects the functional and aesthetic properties of additively manufactured components. Surface roughness can impact the performance, reliability, and durability of the components, as well as their appearance and feel. Surface roughness is an important parameter that affects the structural integrity of additively manufactured (AM) specimens in several ways. Structural integrity refers to the ability of a component to maintain its designed function and structural performance without failure under the service conditions to which it is subjected. Surface roughness, which is the unevenness or irregularities present on the surface of a component, can have a considerable impact on structural integrity. High surface roughness can increase friction and wear, leading to the component's reduced performance and decreased lifespan. Conversely, low surface roughness can improve lubrication and reduce wear, leading to the component's improved performance and increased lifespan. Surface roughness can also affect the reliability of additively manufactured components [21–24]. High surface roughness can create stress concentration points that can lead to cracks and fractures, while low surface roughness can reduce stress concentration and improve the reliability of the component. Surface roughness can also impact the functionality of additively manufactured components. For example, components with low surface roughness may be easier to clean or may be less likely to accumulate dirt and debris, leading to improved functionality and hygiene.

Li et al. [25] employed a data-driven predictive modeling approach to forecast surface roughness in additive manufacturing. The study employed several machine learning algorithms, including Random Forest, AdaBoost, Classification and Regression Trees (CART), Support Vector Regression (SVR), Ridge Regression (RR), and a Random Vector Functional Link (RVFL) network. Wu et al. [26] proposed a novel data-driven approach for surface roughness prediction in Fused Deposition Modeling (FDM). The study aimed to develop an accurate and reliable predictive model that can assist in controlling the quality of the final FDM products. So et al. [27] developed a methodology to enhance the quality of additive manufacturing (AM) products based on data analysis. The study utilized various data analysis techniques, including data pre-processing and Deep Neural Networks (DNNs), combined with sensor data to predict surface roughness. Ulkir et al. [28] conducted a study to determine the optimal combination of input parameters that could predict and minimize surface roughness in Fused Deposition Modeling (FDM) samples produced with a 3D printer. The study utilized a combination of a Cascade-Forward Neural Network (CFNN) and a genetic algorithm to optimize the prediction model.

In the present work, nine supervised machine learning regression-based algorithms are implemented to predict the surface roughness of additively manufactured PLA specimens, with the novel integration of explainable AI techniques to enhance the interpretability and understanding of the model predictions.

## 2. Materials and Methods

To ensure consistency in the model, the ASTM E8 standard geometry was adopted as the reference with a uniform 50% reduction in dimensions to reduce the print size and minimize material usage and time. The response surface methodology (RSM) design of the experiment was employed to generate 30 different trial conditions (refer to Figure 1), each with 3 input parameter levels. Other factors such as material color and color percentage were neglected as these factors have a negligible effect on surface roughness [29]. The CAD model (refer to Figure 2) was sliced using Ultimaker Cura software to generate G-code. A Creality 3D FDM printer (refer to Figure 3) was used to carry out the experimental investigation. Each print was assigned a unique set of settings varying in layer height, infill density, infill pattern, bed temperature, and nozzle temperature to fabricate polylactic acid (PLA) specimens. An input parameter datasheet was created, and the differences in length between each model and the original CAD file were measured using a digital vernier caliper. Using a Mitutoyo SJ-10, surface roughness tester measurements were taken at four locations, and their average was considered.

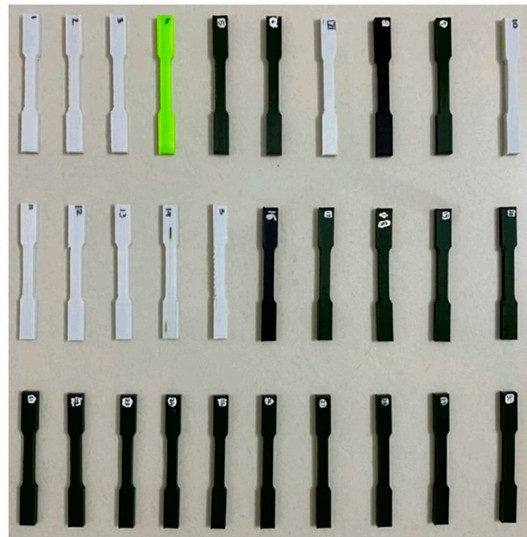

**Figure 1.** Thirty specimens fabricated in the present work.

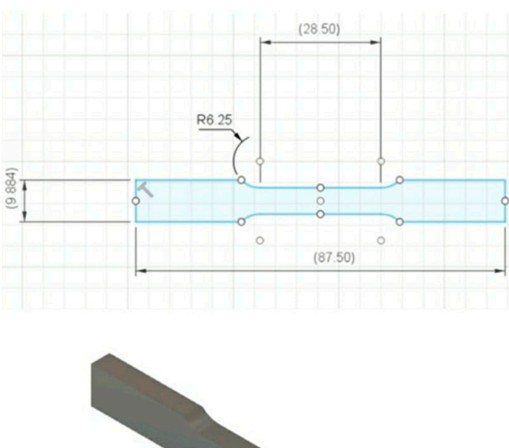

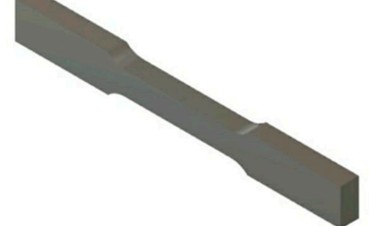

**Figure 2.** Design of the additively manufactured specimens.

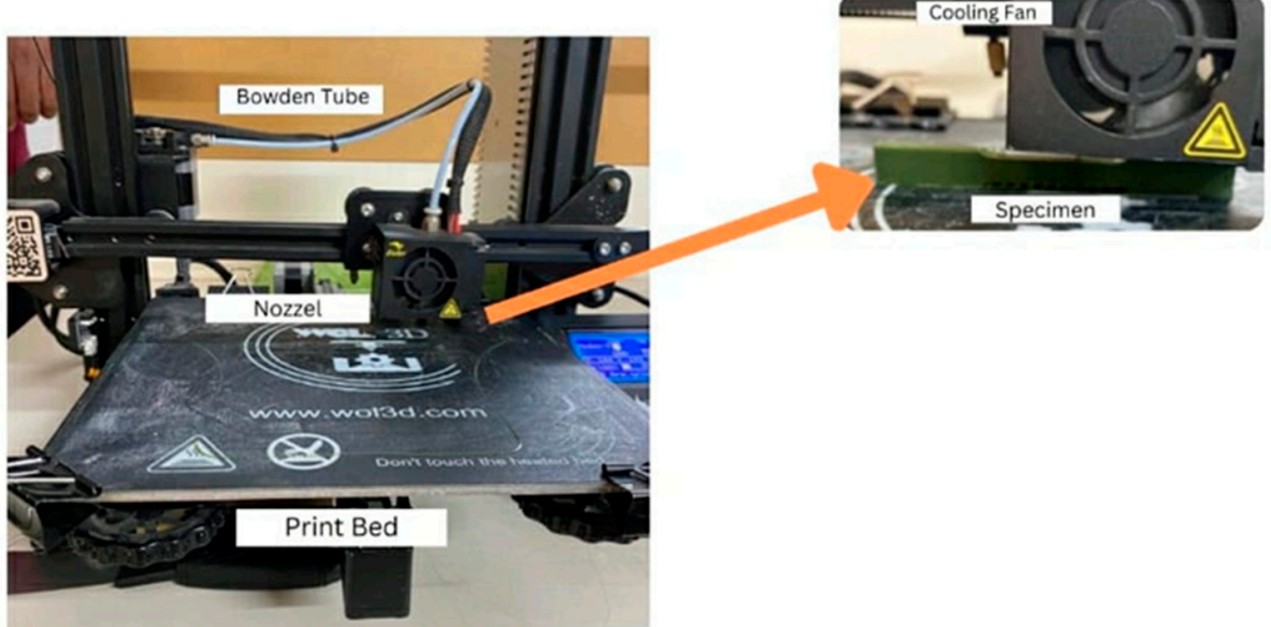

**Figure 3.** Setup for 3D printing the specimens.

The obtained experimental data were converted into a CSV file and were further imported into a Google Colab platform for deploying supervised machine learning regression-based algorithms developed using Python programming. Figure 4 shows the implemented framework in the present work. The Pandas library was used for data manipulation and analysis. It provides data structures for efficiently storing and accessing large datasets. Pandas is widely used in machine learning for tasks such as data pre-processing, cleaning, and transformation. It allows the handling of missing values, merging and grouping of datasets, and filtering and sorting of data. NumPy is used extensively in machine learning. It provides support for large multi-dimensional arrays and matrices, along with a collection of high-level mathematical functions. NumPy is useful in machine learning for tasks such as linear algebra, numerical computing, and scientific computing. The Seaborn library was used for data visualization. It provides a high-level interface for creating attractive and informative statistical graphics. Seaborn is useful in machine learning for visualizing data distributions, detecting patterns, and exploring relationships between variables. Matplotlib is another Python library used for data visualization. It is a comprehensive library that provides a wide range of graphical tools for creating high-quality visualizations. Matplotlib is useful in machine learning for tasks such as data visualization, model evaluation, and result presentation.

In machine learning, a critical aspect of building a model is evaluating its performance. To do this, data scientists divide the dataset into two parts: training data and testing data. Typically, 80% of the dataset is used for training, while the remaining 20% is used for testing. The primary reason for using an 80/20 split is to reduce overfitting, which occurs when a model is too complex and fits the training data too closely. By using a smaller portion of the dataset for testing, data scientists can ensure that the model generalizes well to new data.

In the present work, mean absolute error (MAE), mean square error (MSE), and coefficient of determination ($R^2$) were the metric features used for measuring the performance of the machine learning models.

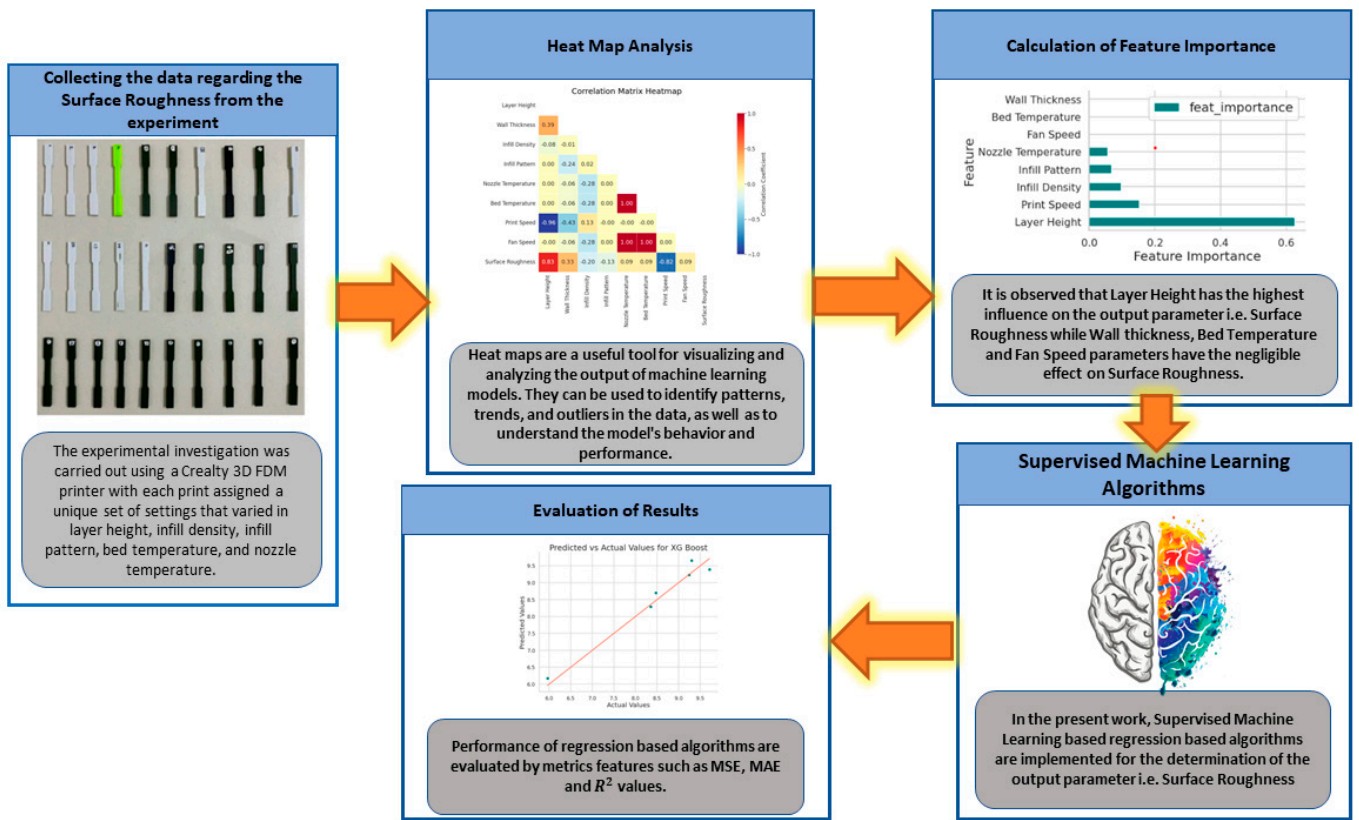

**Figure 4.** Implemented machine learning framework in the present work.

## 3. Results

Table 1 shows the obtained results for surface roughness according to the combination of different input parameters.

**Table 1.** Experimental results.

| Layer Height (mm) | Wall Thickness (mm) | Infill Density (%) | Infill Pattern | Nozzle Temperature (°C) | Bed Temperature (°C) | Print Speed (mm/sec) | Fan Speed (%) | Surface Roughness (μm) |
|---|---|---|---|---|---|---|---|---|
| 0.1 | 1 | 50 | Honeycomb | 200 | 60 | 120 | 0 | 6.12275 |
| 0.1 | 4 | 40 | grid | 205 | 65 | 120 | 25 | 6.35675 |
| 0.1 | 3 | 50 | Honeycomb | 210 | 70 | 120 | 50 | 5.957 |
| 0.1 | 4 | 90 | Grid | 215 | 75 | 120 | 75 | 5.92025 |
| 0.1 | 1 | 30 | Honeycomb | 220 | 80 | 120 | 100 | 6.08775 |
| 0.15 | 3 | 80 | Honeycomb | 200 | 60 | 60 | 0 | 6.0684 |
| 0.15 | 4 | 50 | Grid | 205 | 65 | 60 | 25 | 9.27525 |
| 0.15 | 10 | 30 | Honeycomb | 210 | 70 | 60 | 50 | 7.479 |
| 0.15 | 6 | 40 | Grid | 215 | 75 | 60 | 75 | 7.557 |
| 0.15 | 1 | 10 | Honeycomb | 220 | 80 | 60 | 100 | 8.48675 |
| 0.2 | 5 | 60 | Honeycomb | 200 | 60 | 40 | 0 | 8.4695 |
| 0.2 | 4 | 20 | Grid | 205 | 65 | 40 | 25 | 8.8785 |
| 0.2 | 5 | 60 | Honeycomb | 210 | 70 | 40 | 50 | 9.415 |

**Table 1.** *Cont.*

| Layer Height (mm) | Wall Thickness (mm) | Infill Density (%) | Infill Pattern | Nozzle Temperature (°C) | Bed Temperature (°C) | Print Speed (mm/sec) | Fan Speed (%) | Surface Roughness (μm) |
|---|---|---|---|---|---|---|---|---|
| 0.2 | 7 | 40 | Grid | 215 | 75 | 40 | 75 | 9.71375 |
| 0.2 | 3 | 60 | Honeycomb | 220 | 80 | 40 | 100 | 10.59625 |
| 0.1 | 1 | 50 | Triangles | 200 | 60 | 120 | 0 | 6.04925 |
| 0.1 | 4 | 40 | Cubic | 205 | 65 | 120 | 25 | 9.262 |
| 0.1 | 3 | 50 | Triangles | 210 | 70 | 120 | 50 | 6.127 |
| 0.1 | 4 | 90 | Cubic | 215 | 75 | 120 | 75 | 5.99675 |
| 0.1 | 1 | 30 | Triangles | 220 | 80 | 120 | 100 | 6.1485 |
| 0.15 | 3 | 80 | Triangles | 200 | 60 | 60 | 0 | 8.2585 |
| 0.15 | 4 | 50 | Cubic | 205 | 65 | 60 | 25 | 8.347 |
| 0.15 | 10 | 30 | Triangles | 210 | 70 | 60 | 50 | 8.2385 |
| 0.15 | 6 | 40 | Cubic | 215 | 75 | 60 | 75 | 8.23125 |
| 0.15 | 1 | 10 | Triangles | 220 | 80 | 60 | 100 | 8.35125 |
| 0.2 | 5 | 60 | Triangles | 200 | 60 | 40 | 0 | 9.072 |
| 0.2 | 4 | 20 | Cubic | 205 | 65 | 40 | 25 | 9.23825 |
| 0.2 | 5 | 60 | Triangles | 210 | 70 | 40 | 50 | 9.18225 |
| 0.2 | 7 | 40 | Cubic | 215 | 75 | 40 | 75 | 9.299 |
| 0.2 | 3 | 60 | Triangles | 220 | 80 | 40 | 100 | 9.382 |

## 3.1. Supervised Machine Learning Algorithms

Figure 5 shows the obtained correlation heat map matrix in the present work. A correlation matrix heatmap is an essential tool in machine learning because it helps to identify the strength and direction of the relationship between different variables. It provides a quick visual representation of how different variables are related to each other. This information is critical in feature selection, as highly correlated variables can lead to overfitting, and it is important to remove redundant variables to improve a model's performance. The correlation matrix heatmap is color-coded, with the intensity of the color representing the strength of the correlation. A positive correlation is represented as a shade of red, while a negative correlation is represented as a shade of blue. The darker the shade, the stronger the correlation. A neutral correlation is represented as a shade of white or gray. Variables that are highly correlated with each other appear as dark squares on the heatmap. These variables can lead to overfitting and should be removed. Variables that have little or no correlation with each other appear as light squares on the heatmap. A positive correlation between two variables is represented as a shade of red. If the variables have a strong positive correlation, it means that they move in the same direction. A negative correlation between two variables is represented as a shade of blue. If variables have a strong negative correlation, it means that they move in opposite directions.

### 3.1.1. Decision Tree Algorithm

Decision Tree Regression is a prominent non-parametric machine learning approach that establishes a relationship between input features (X) and a continuous target variable (Y) by iteratively partitioning the input space into distinct subsets. The primary objective of this partitioning process is to minimize the impurity within each node [30,31]. Commonly employed criteria for measuring impurity include mean squared error (MSE), mean ab-

solute error (MAE), and coefficient of determination ($R^2$). The impurity of a node can be computed using Equation (1) for the MSE.

$$MSE = \frac{\sum(y_i - \hat{y})^2}{N}$$ (1)

where N denotes the number of samples in the node, $y_i$ represents the actual target value for the i-th sample, and $\hat{y}$ is the average target value of the samples in the node.

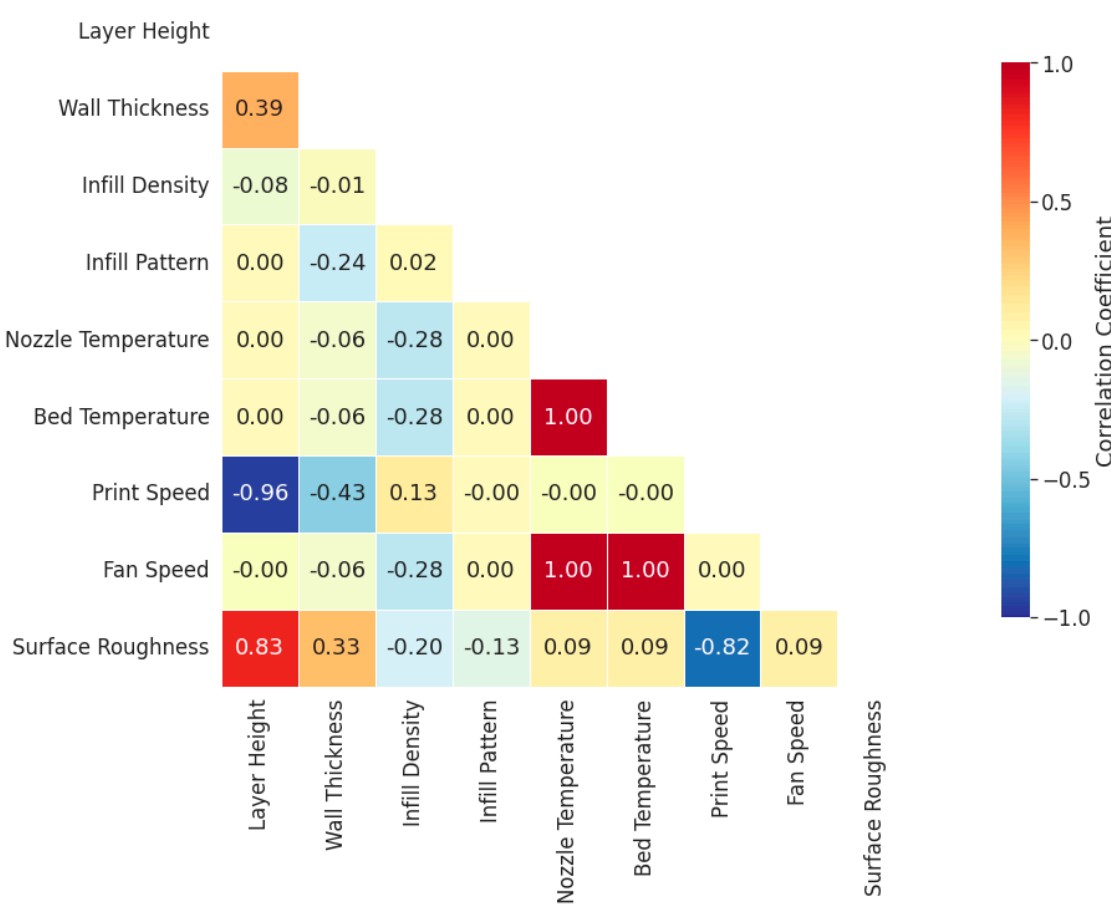

**Figure 5.** Correlation matrix heatmap plot.

Commencing from the root node, the input space is iteratively divided into subsets by identifying the optimal split that minimizes the selected impurity criterion. This procedure persists until a stopping condition is achieved, such as reaching a maximum depth, attaining a minimum number of samples in a leaf node, or meeting an impurity threshold. Subsequent to partitioning, the average target value of the samples in each leaf node serves as the prediction for the corresponding region of the input space. This results in a piecewise constant function that models the relationship between X and Y.

Figure 6 shows the feature importance plot obtained in the present work. A feature importance plot is a visual tool used in machine learning to determine the importance of each feature in a dataset. It helps in identifying which features are most relevant to the target variable and which can be eliminated. A feature importance plot can also help in identifying irrelevant features that do not contribute to a model's accuracy. These features can be eliminated during feature selection to improve the model's performance. It is observed that layer height has the highest influence on the output parameter, i.e., surface roughness, while the wall thickness, bed temperature, and fan speed parameters have negligible effects on surface roughness.

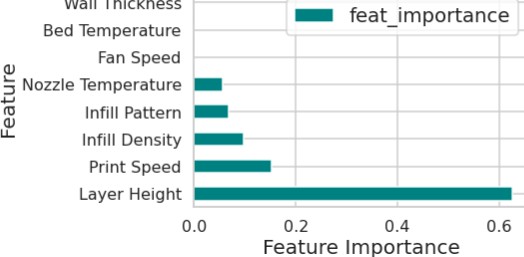

**Figure 6.** Feature importance plot.

Figure 7 shows the obtained Decision Tree plot. Decision Tree plots are crucial in machine learning as they provide a visual representation of the decision-making process used by a model. These plots show the hierarchy of decision rules used to classify or predict outcomes and help in interpreting the model's behavior. The information obtained from a Decision Tree plot can help in understanding the important features that contribute to the model's accuracy and identify areas where the model can be improved. Additionally, Decision Tree plots can aid in explaining the model's predictions to non-technical stakeholders, making it an essential tool for both model developers and end-users. Therefore, Decision Tree plots are an important component of the machine learning toolkit and can provide valuable insights into the underlying decision-making process of a model.

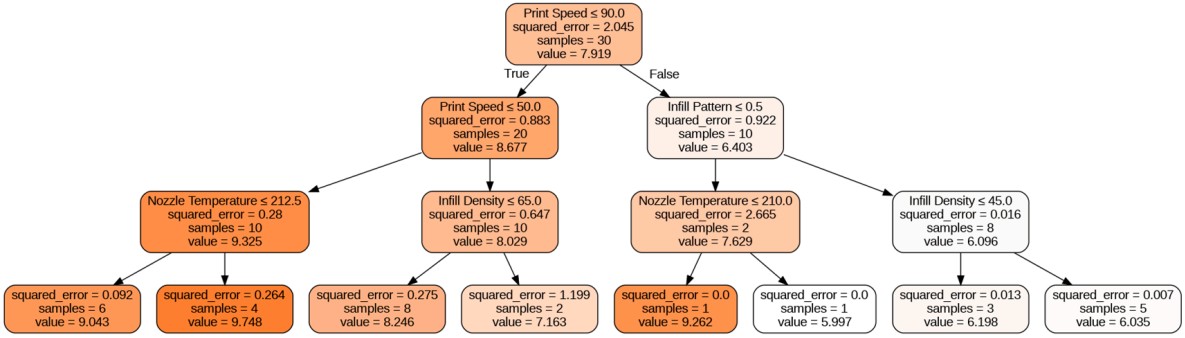

**Figure 7.** Decision Tree plot.

Figure 8 shows the plot obtained between the predicted values and the original values from the implemented machine learning algorithms. Table 2 shows the comparison of the performance of the implemented supervised machine learning regression-based algorithms on the basis of metric features such as MSE, MAE, and $R^2$ value.

**Table 2.** Evaluating the performance of supervised machine learning algorithms to predict surface roughness.

| Algorithm | MSE | MAE | $R^2$ Value |
|---|---|---|---|
| Support Vector Regression | 0.269872 | 0.460177 | 0.822964 |
| Decision Tree | 0.498258 | 0.568042 | 0.673144 |
| Random Forest | 0.123575 | 0.295479 | 0.918935 |
| XGBoost | 0.055762 | 0.199178 | 0.963421 |
| CatBoost | 0.561542 | 0.657891 | 0.631630 |
| AdaBoost | 0.161093 | 0.329904 | 0.894323 |
| Extra Tree Regressor | 0.142213 | 0.304175 | 0.906709 |
| Gradient Boosting Regressor | 0.144944 | 0.317427 | 0.904917 |
| Explainable Boosting Model (EBM) | 0.343811 | 0.554507 | 0.750395 |

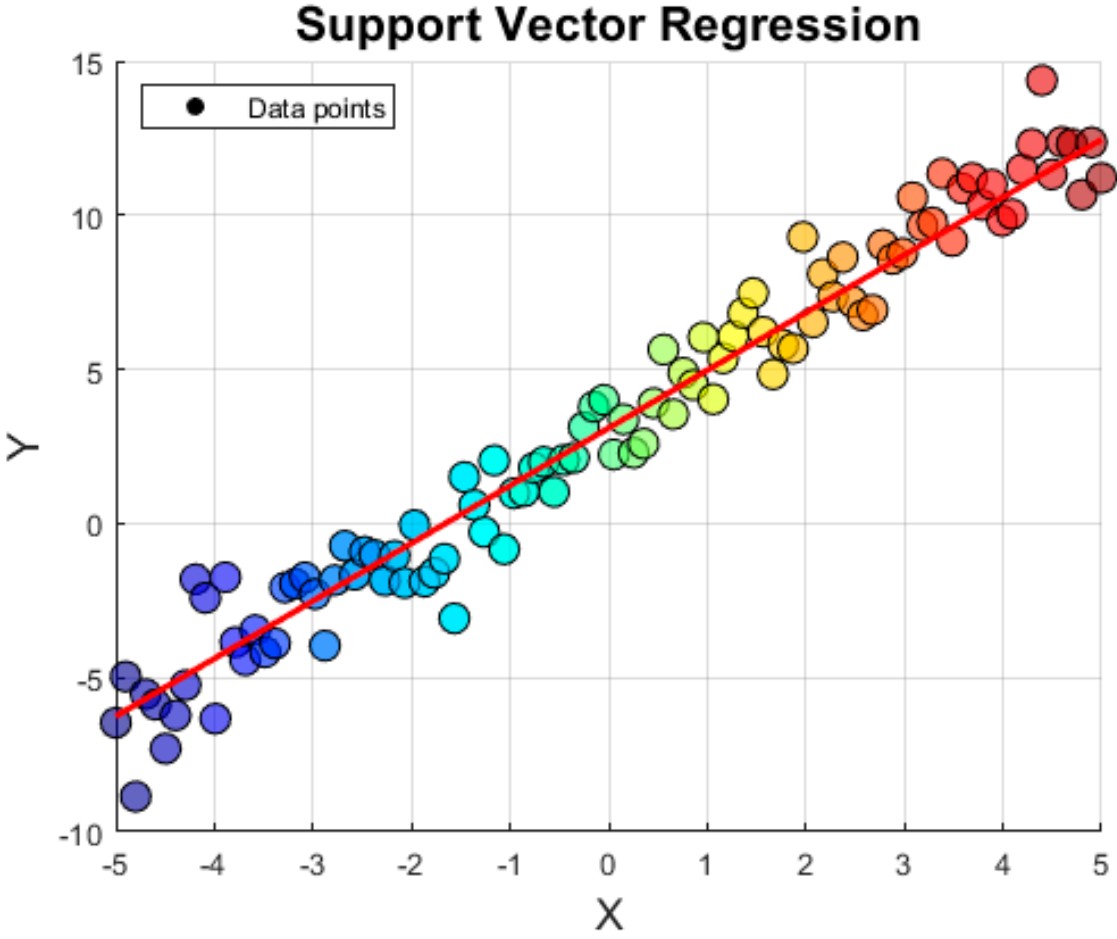

**Figure 8.** Schematic representation of SVR model's prediction.

3.1.2. Support Vector Regression

Support Vector Regression (SVR) is an advanced machine learning technique employed for regression tasks, focusing on modeling the relationship between input features and a continuous target variable. The principal aim of SVR is to discover a function f(x) that approximates the target variable, y, within a predefined margin of tolerance ($\varepsilon$) [32]. By formulating an $\varepsilon$-insensitive loss function, SVR demonstrates robustness against noise and outliers present in the data. Mathematically, the objective is to minimize the objective function shown in Equation (2).

$$L(w, b) = \frac{1}{2}\|w\|^2 + C \times \Sigma(\xi_i + \xi^*_i) \tag{2}$$

subjected to the following constraints:

$$y_i - f(x_i) \leq \varepsilon + \xi_i,$$

$$f(x_i) - y_i \leq \varepsilon + \xi_i,$$

$$\xi, \xi_i \geq 0,$$

where $x_i$ and $y_i$ represent the input features and target values, respectively, w denotes the weight vector, b corresponds to the bias term, $\xi_i$ and $\xi_i$ are the slack variables for the upper and lower bounds, respectively, and C is the regularization parameter that balances the trade-off between minimizing model complexity ($\|w\|^2$) and permitting a certain level of error ($\xi_i + \xi_i$) within the $\varepsilon$-margin.

The plot shown in Figure 8 illustrates the original data points as a vibrant scatter plot and the SVR model's predictions as a red line. By minimizing the objective function and

adhering to the constraints, the SVR model effectively captures the underlying relationship between the input features and the continuous target variable.

### 3.1.3. Random Forest

Random Forest is a powerful ensemble learning method employed in regression and classification tasks. It amalgamates multiple Decision Trees to produce more accurate and stable predictions. The core principle behind Random Forest is the exploitation of collective wisdom by averaging predictions from numerous weak learners (Decision Trees) to construct a more reliable and precise model [33].

In the Random Forest algorithm, each Decision Tree is built utilizing a random subset of the training data, sampled with replacement (bootstrap sampling). Additionally, at each node, a random subset of features is considered for partitioning. This randomization process in both data and feature selection mitigates correlation among individual trees and augments the diversity within the ensemble.

For regression tasks, the Random Forest model's prediction is computed as the mean of the predictions from all the constituent trees in the ensemble as shown in Equation (3).

$$f(x) = \frac{\sum f_i(x)}{N} \tag{3}$$

where $f_i(x)$ represents the prediction of the i-th Decision Tree for input x, and N denotes the total number of trees in the ensemble.

During the training phase, each Decision Tree grows independently to its maximum depth or until a stopping criterion is fulfilled, such as reaching a minimum number of samples in a leaf node. Commonly employed impurity measures for regression tasks include mean squared error (MSE) and mean absolute error (MAE). Pruning is unnecessary in the context of Random Forests, as the averaging process inherently counteracts overfitting that might transpire in individual trees.

The performance of a Random Forest model can be evaluated using out-of-bag (OOB) error estimation. As each tree is trained on a random data subset, approximately one-third of the samples remain unused for training (out-of-bag samples). These samples can be leveraged to estimate the model's performance on unseen data, eliminating the need for cross-validation.

### 3.1.4. XGBoost

XGBoost, an acronym for eXtreme gradient boosting, is a sophisticated ensemble learning method employed in both regression and classification tasks. It represents an optimized and scalable variant of gradient boosting, an approach that assembles a robust model by fitting weak learners sequentially (commonly Decision Trees) to minimize a differentiable loss function [34].

The fundamental concept underlying gradient boosting involves fitting each weak learner to the negative gradient of the loss function relative to the predictions of preceding learners. This iterative process refines the model by successively reducing residual errors. The final model is constituted by a weighted sum of the predictions from all weak learners, as shown in Equation (4).

$$f(x) = \sum \alpha_i \cdot f_i(x) \tag{4}$$

where $f_i(x)$ denotes the prediction of the i-th weak learner for input x, and $\alpha_i$ signifies the weight attributed to that learner.

During the training phase, the weights ($\alpha_i$) are determined by minimizing the loss function. For differentiable loss functions, this procedure can be viewed as gradient descent within the function space. In every iteration, the algorithm fits a weak learner to the negative gradient of the loss function in relation to the current model's predictions, subsequently updating the model by incorporating this new learner with an optimal weight.

### 3.1.5. CatBoost

CatBoost, derived from "Category" and "Boosting," is a state-of-the-art ensemble learning technique devised to manage categorical variables in regression and classification tasks. It constitutes a highly optimized implementation of gradient boosting that assembles a robust model by fitting weak learners (typically Decision Trees) sequentially to minimize a differentiable loss function [35].

The fundamental concept underlying gradient boosting involves fitting each weak learner to the negative gradient of the loss function concerning the predictions of previous learners. Through this iterative process, the model is refined by successively reducing residual errors.

A key feature of CatBoost is its capacity to efficiently handle categorical variables through an innovative encoding scheme, termed "ordered boosting." This method computes the target mean for each categorical feature level based on historical data, minimizing the risk of target leakage. Mathematically, the encoding for the i-th level of a categorical variable is expressed as Equation (5).

$$\mu_i = \frac{\sum y_j}{n_i} \tag{5}$$

where $y_j$ denotes the target values corresponding to the i-th level of the categorical variable, and $n_i$ signifies the number of occurrences of the i-th level in the data.

### 3.1.6. AdaBoost

AdaBoost, an abbreviation for adaptive boosting, is a prominent ensemble learning approach employed in both regression and classification tasks. It functions by iteratively fitting weak learners (commonly decision stumps or shallow Decision Trees) to the training data while dynamically adjusting the weights of instances to prioritize misclassified samples [36].

The fundamental concept underlying AdaBoost involves assigning equal initial weights to each instance in the training data. During each iteration, the algorithm fits a weak learner to the weighted data, concentrating on minimizing the weighted error.

Following each iteration, the instance weights are updated to emphasize misclassified samples, guiding the subsequent weak learner to focus on more challenging instances.

Mathematically, the weight update rule is expressed as Equation (6).

$$w_j = w_j \cdot e^{(\alpha_i \cdot I(y_j \neq f_i(x_j)))} \tag{6}$$

where $w_j$ denotes the weight of the j-th instance, $y_j$ corresponds to its true label, $x_j$ refers to the input feature vector, I() is an indicator function that assumes a value of one when the condition within the parenthesis is true and zero otherwise.

### 3.1.7. Extra Tree Regressor

The Extra Tree Regressor, an abbreviation for Extremely Randomized Trees Regressor, is an ensemble learning technique utilized for regression tasks. It is an extension of the Random Forest algorithm, focusing on the construction of Decision Trees with additional randomization during the tree-building process. This randomization helps create more diverse and independent trees, reducing the overall model variance and improving generalization performance [37].

The core principle of the Extra Trees Regressor involves constructing multiple Decision Trees using a random subset of features and samples from the training data. While the Random Forest algorithm selects the optimal split points based on a criterion such as the mean squared error (MSE) or the Gini impurity, the Extra Trees Regressor introduces extra randomization by choosing split points randomly for each feature within a randomly selected subset. This process is applied to each node in the tree during its construction.

### 3.1.8. Gradient Boosting Regressor

The Gradient Boosting Regressor is a powerful ensemble learning technique employed for regression tasks. It builds a robust model by iteratively fitting weak learners (typically shallow Decision Trees) to the training data, with each learner focusing on reducing the residual errors of the preceding learners. This method is founded on the principles of gradient boosting, which involves optimizing a differentiable loss function through the iterative refinement of the model [38]. The core concept of the Gradient Boosting Regressor is to fit each weak learner to the negative gradient of the loss function concerning the predictions of the previous learners. This iterative process refines the model by successively reducing residual errors.

### 3.1.9. Explainable Boosting Model

The Explainable Boosting Model (EBM) is a machine learning algorithm that combines multiple one-dimensional models to generate an interpretable and accurate final model. To achieve interpretability, the EBM trains several simple models for each feature and then combines them to produce an additive model. The model is built iteratively by adjusting the weights of each model based on prediction errors [39]. This process enables the EBM to provide a clear understanding of each feature's effect on the target variable. The model's interpretability is demonstrated in its ability to provide both global and local interpretations of the predictions. The global interpretation highlights the overall effect of each feature, while the local interpretation sheds light on how each feature affects a particular data point. The EBM is adaptable to various datasets and offers a powerful and transparent approach to machine learning that is useful for many applications. In summary, the EBM is a highly interpretable machine learning algorithm that delivers accurate predictions while providing a granular understanding of the model's behavior, making it an attractive choice for tasks that require both accuracy and transparency.

Figure 9 shows the plot obtained between the predicted values and the original values from the implemented machine learning algorithms. Table 2 shows the comparison of the performances of the implemented supervised machine learning regression-based algorithms on the basis of metric features such as MSE, MAE, and $R^2$ value represented in Equations (7)–(9).

$$\text{MSE} = \frac{\sum(y_i - \hat{y}_i)^2}{N} \tag{7}$$

$$\text{MAE} = \frac{\sum|y_i - \hat{y}_i|}{N} \tag{8}$$

$$R^2 = 1 - \frac{\sum(y_i - \hat{y}_i)^2}{\sum\left(y_i - \bar{y}\right)^2} \tag{9}$$

where N is the number of samples, $y_i$ is the true value of the i-th sample, $\bar{y}$ is the mean of the true values, and $\hat{y}_i$ is the predicted value of the i-th sample.

### 3.2. Explainable Artificial Intelligence (XAI) Approach

Explainable AI (XAI) aims to provide human-understandable explanations for the decisions made by machine learning models. In our case, we have a set of input parameters that influence the surface roughness of additively manufactured specimens. XAI can be applied to make the relationship between these input parameters and surface roughness more transparent and interpretable.

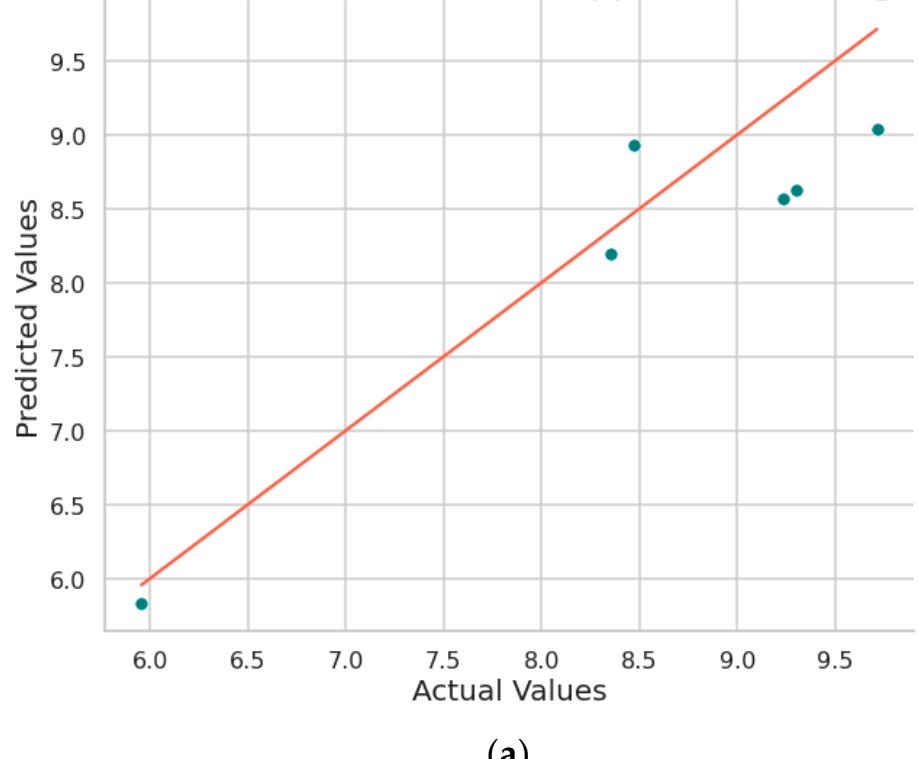

(**a**)

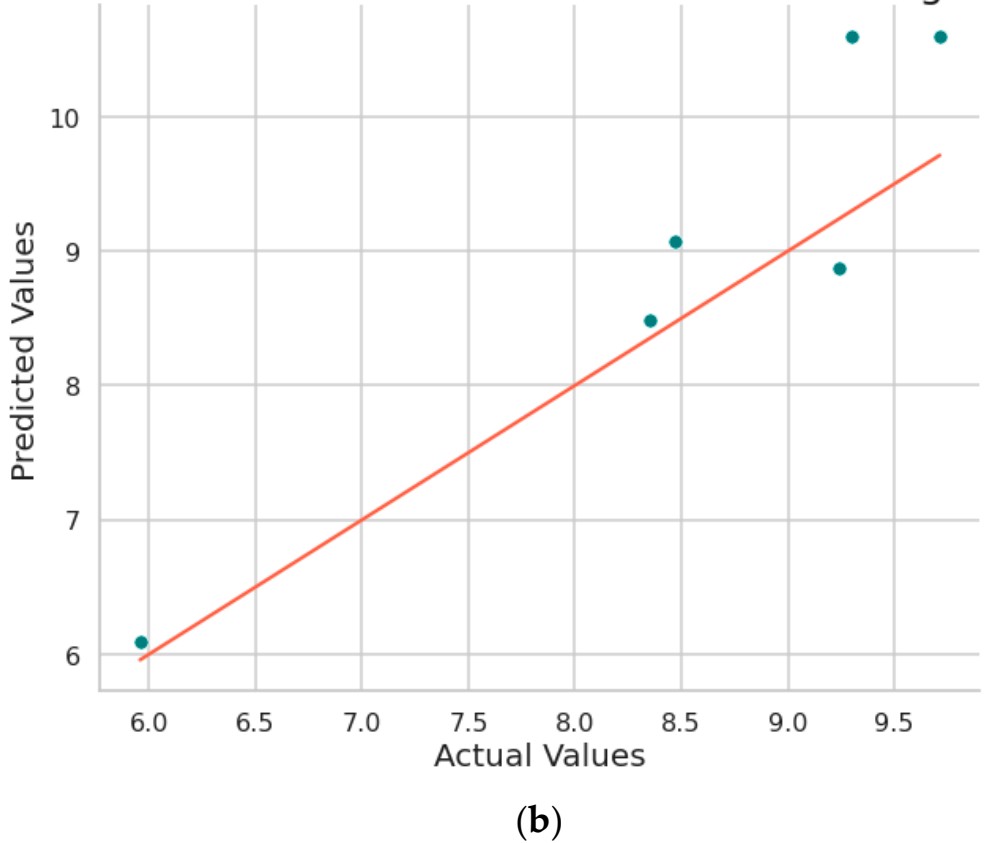

(**b**)

**Figure 9.** *Cont.*

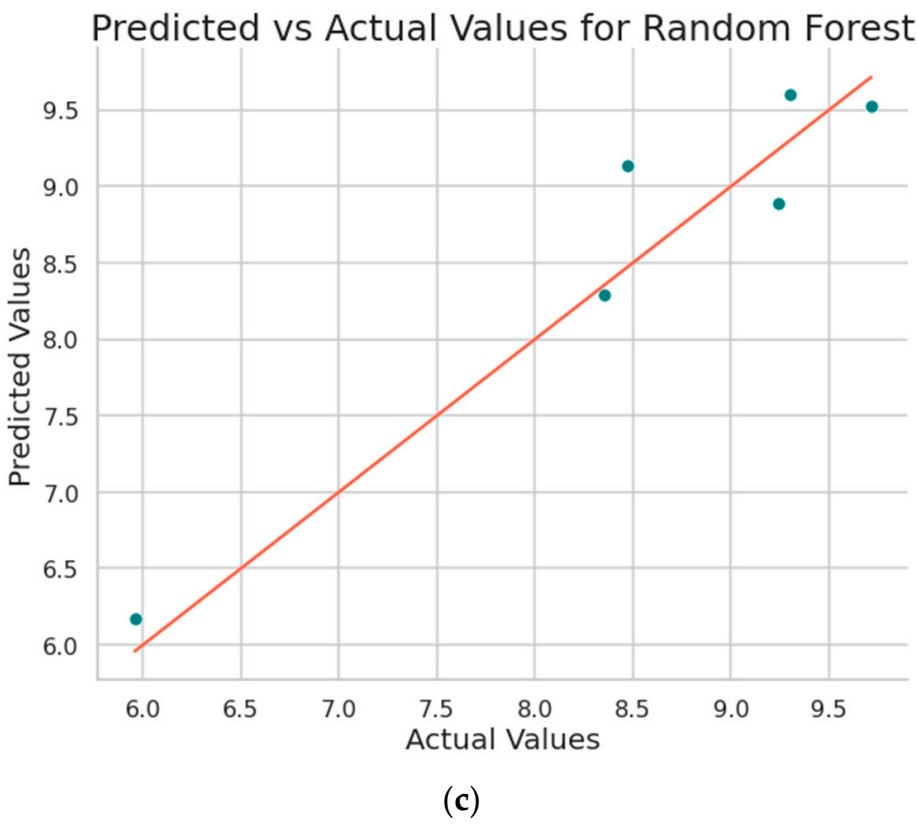

(**c**)

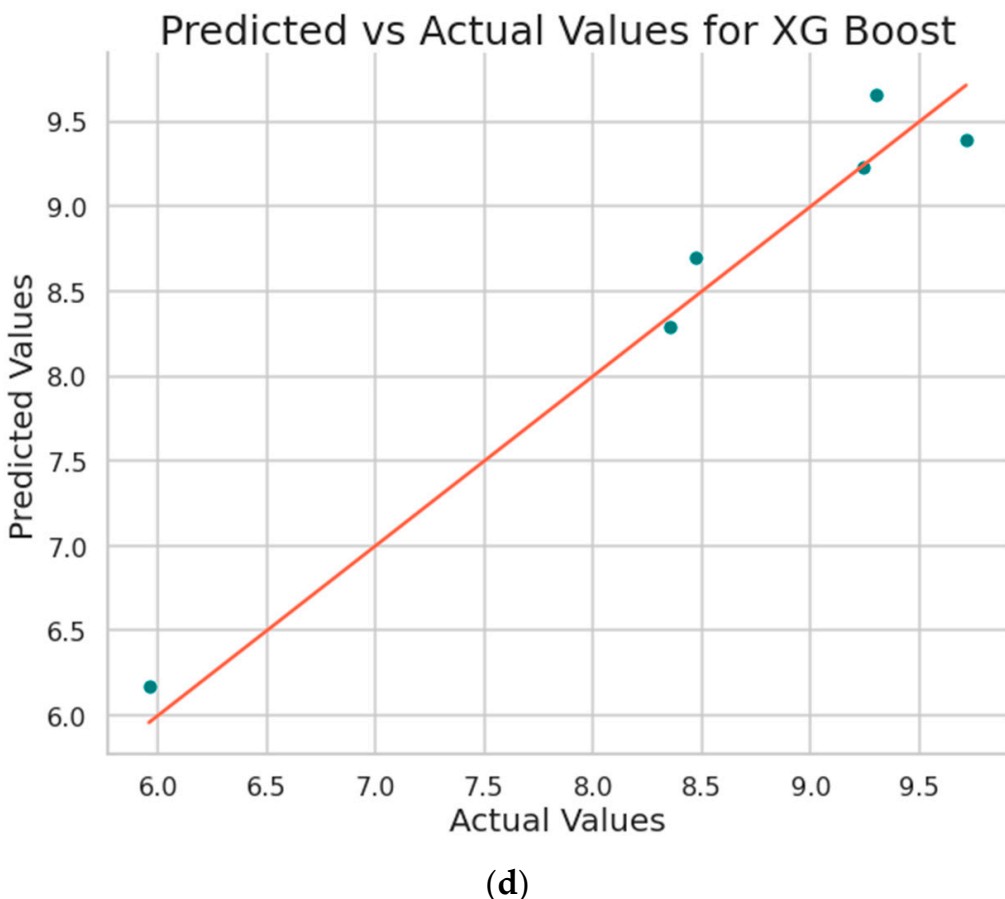

(**d**)

**Figure 9.** *Cont.*

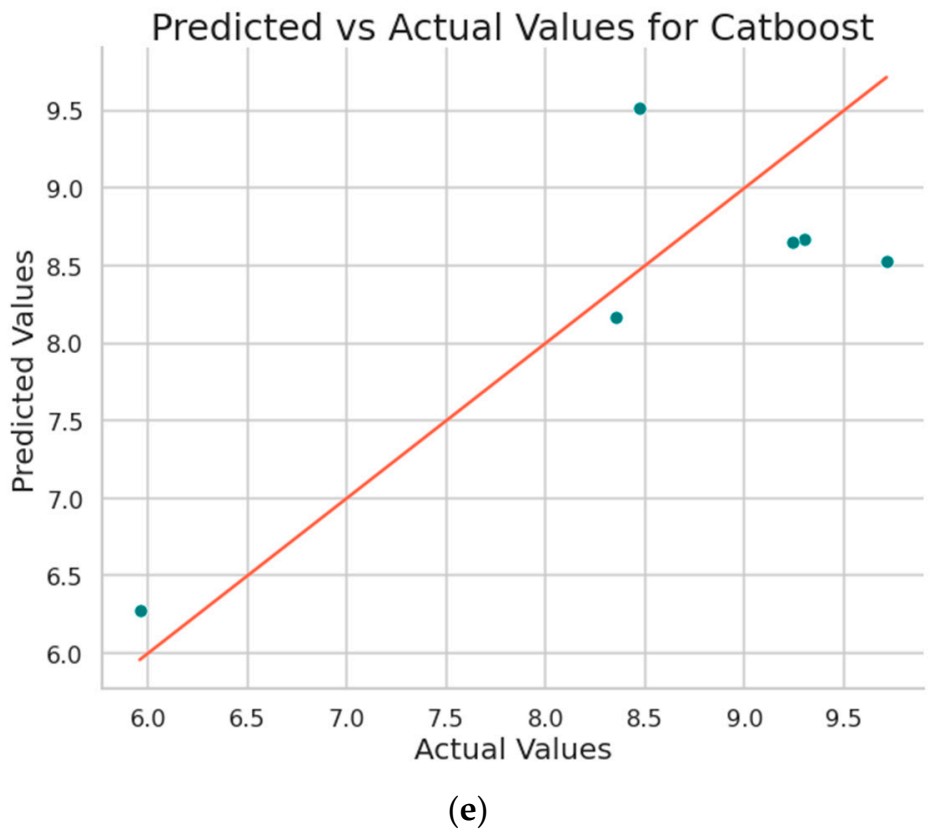

(**e**)

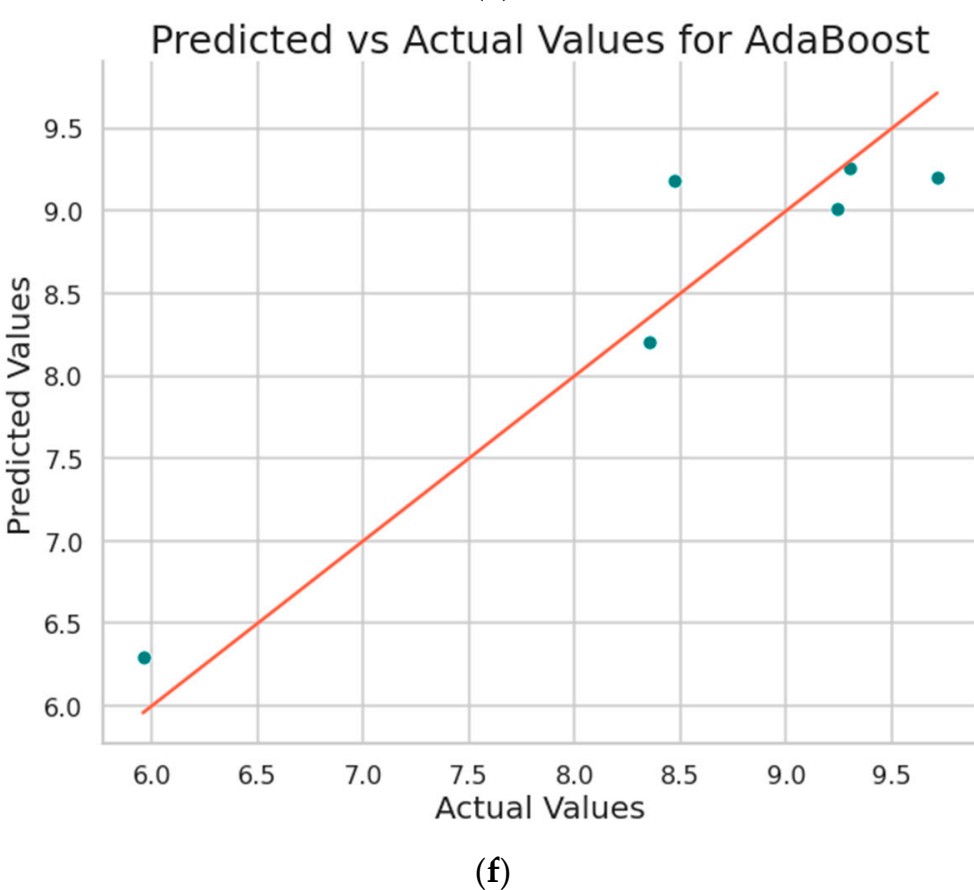

(**f**)

**Figure 9.** *Cont.*

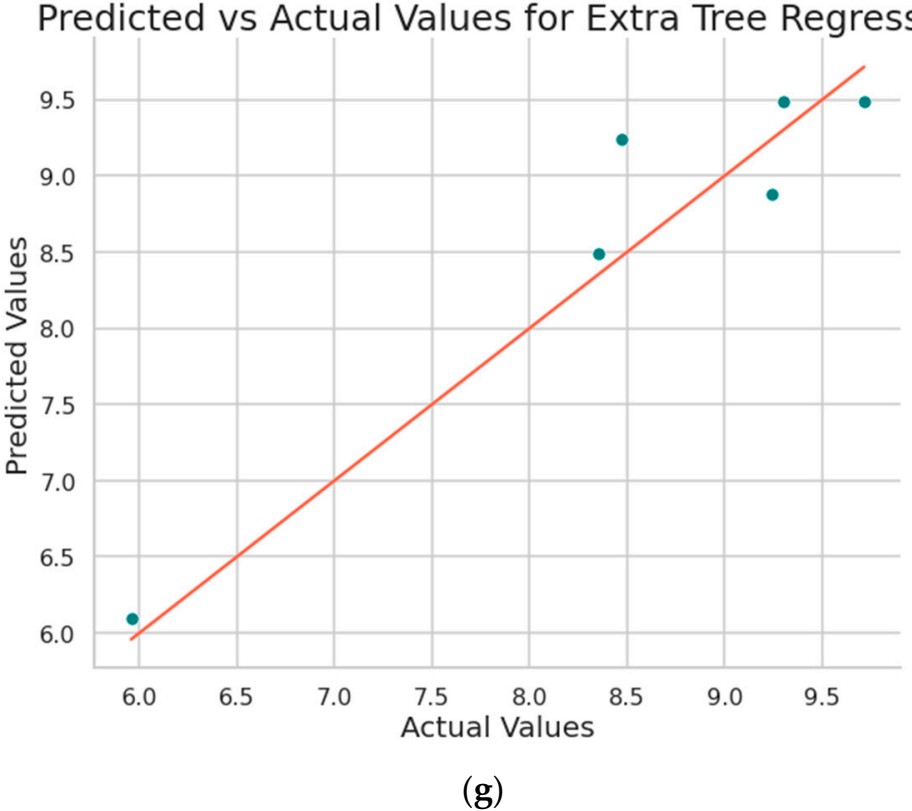

(**g**)

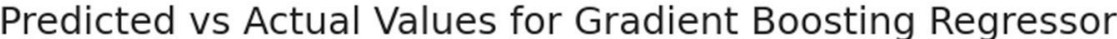

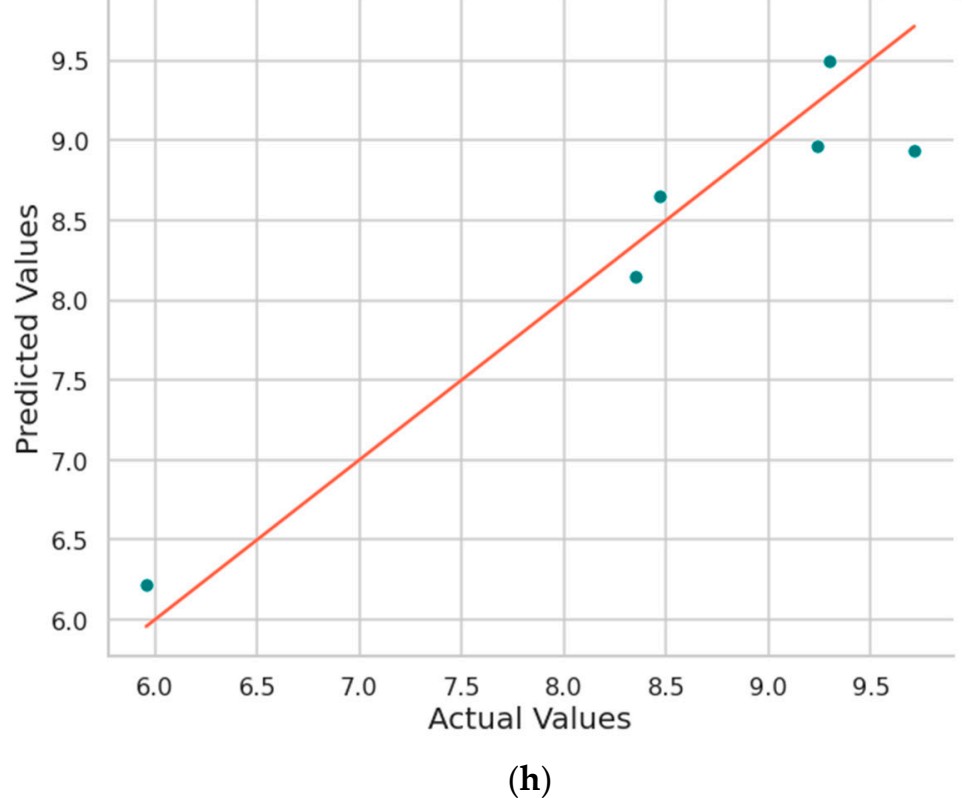

(**h**)

**Figure 9.** *Cont.*

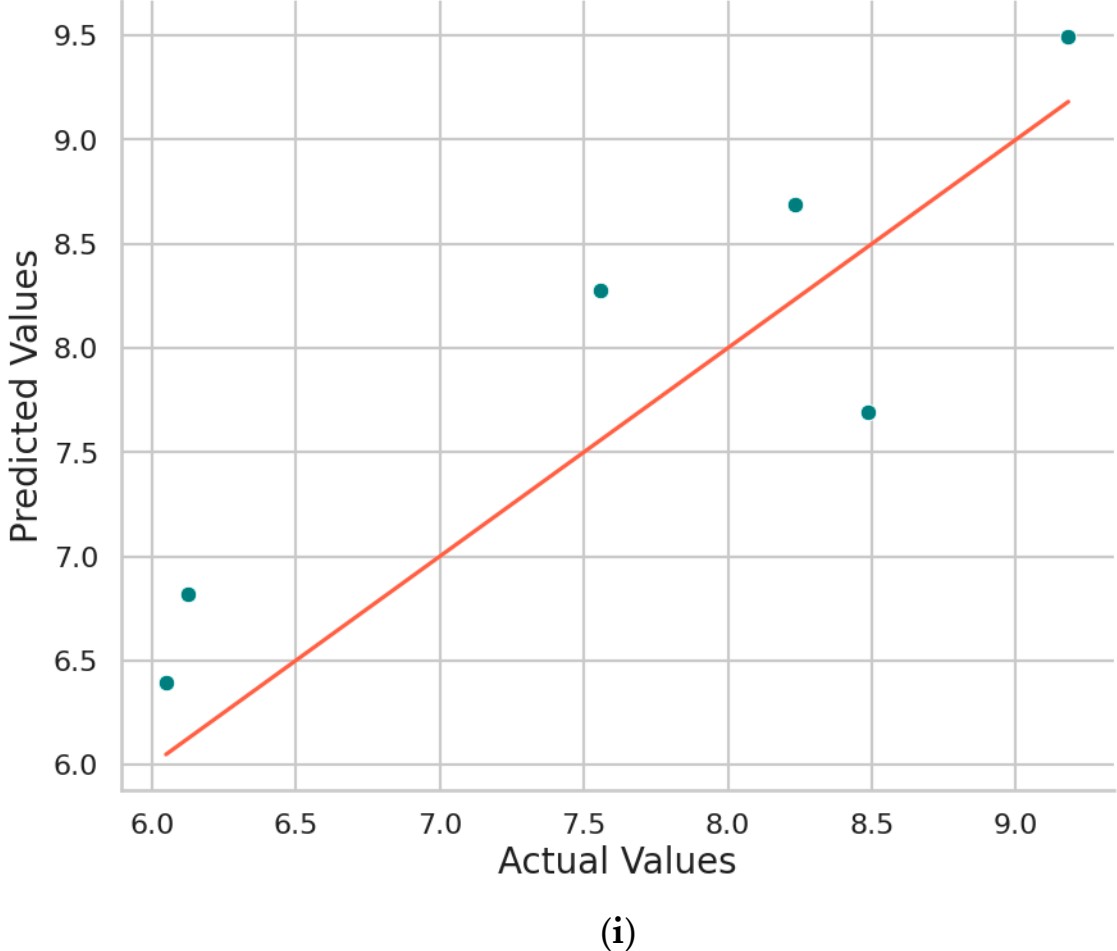

(**i**)

**Figure 9.** Predicted vs. actual surface roughness values plots obtained for (**a**) Support Vector Regression, (**b**) Decision Tree, (**c**) Random Forest, (**d**) XGBoost, (**e**) CatBoost, (**f**) AdaBoost, (**g**) Extra Tree Regressor, (**h**) Gradient Boosting Regressor, and (**i**) Explainable Boosting Model (EBM).

The partial dependence plot created using SHAP values, as shown in Figure 10, provides valuable insights into the relationship between a specific input feature and the model's predictions [40]. This plot can be used to analyze the effect of the chosen feature on the predicted output while accounting for the average influence of all the other features. By interpreting this plot, researchers can gain a deeper understanding of the model's behavior and make informed decisions.

The waterfall plot shown in Figure 11 is a visual representation that helps to understand the step-by-step contributions of individual features to the model's prediction for a specific instance. This plot is useful for interpreting the model's behavior and attributing importance to each feature in a clear, ordered manner.

Figure 12 demonstrates the process of fitting an Explainable Boosting Machine (EBM) model to the data and using SHAP values to create a partial dependence plot for the input features. The EBM is a type of Generalized Additive Model (GAM) that provides interpretable results through additive combinations of simple models.

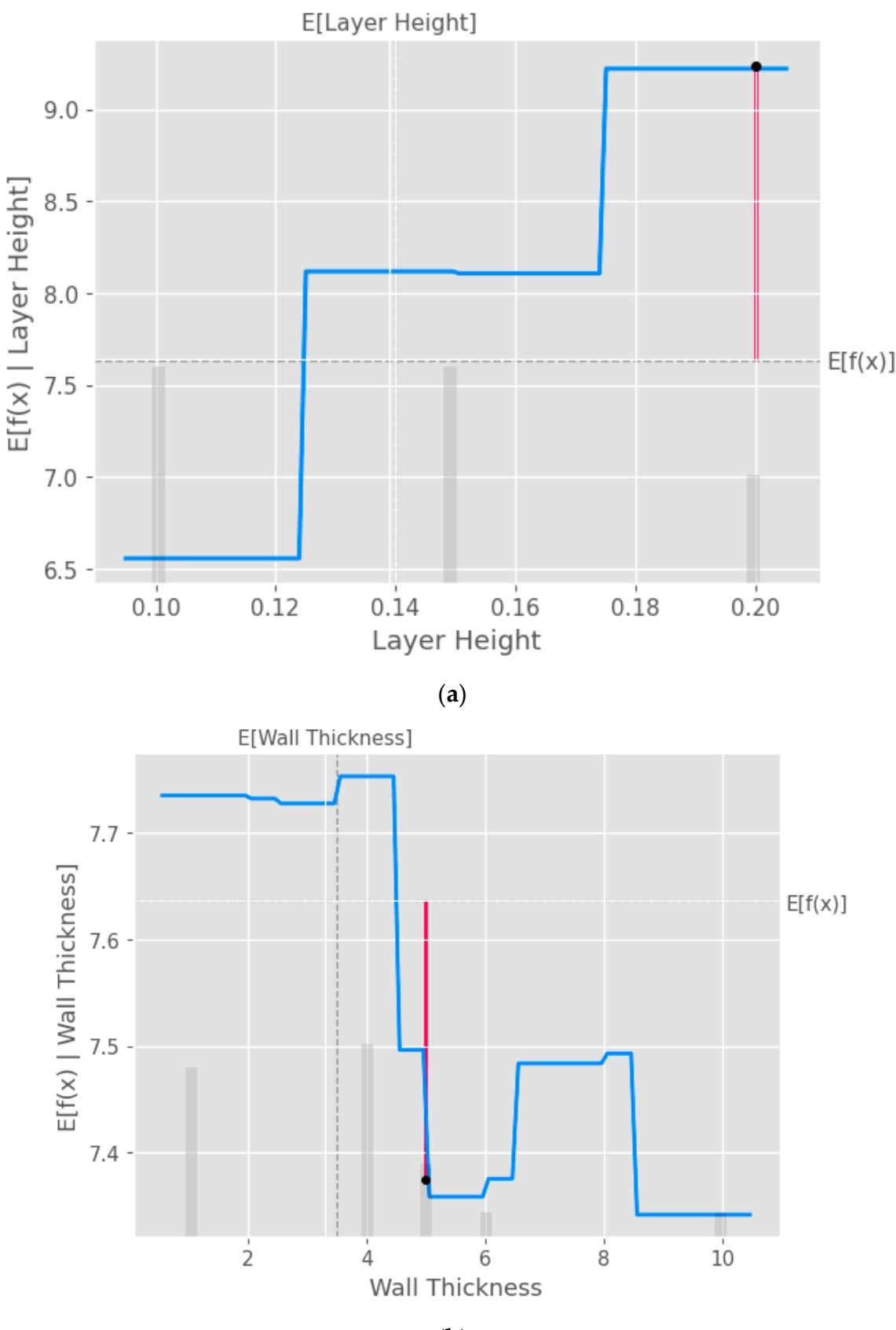

(**a**)

(**b**)

**Figure 10.** *Cont.*

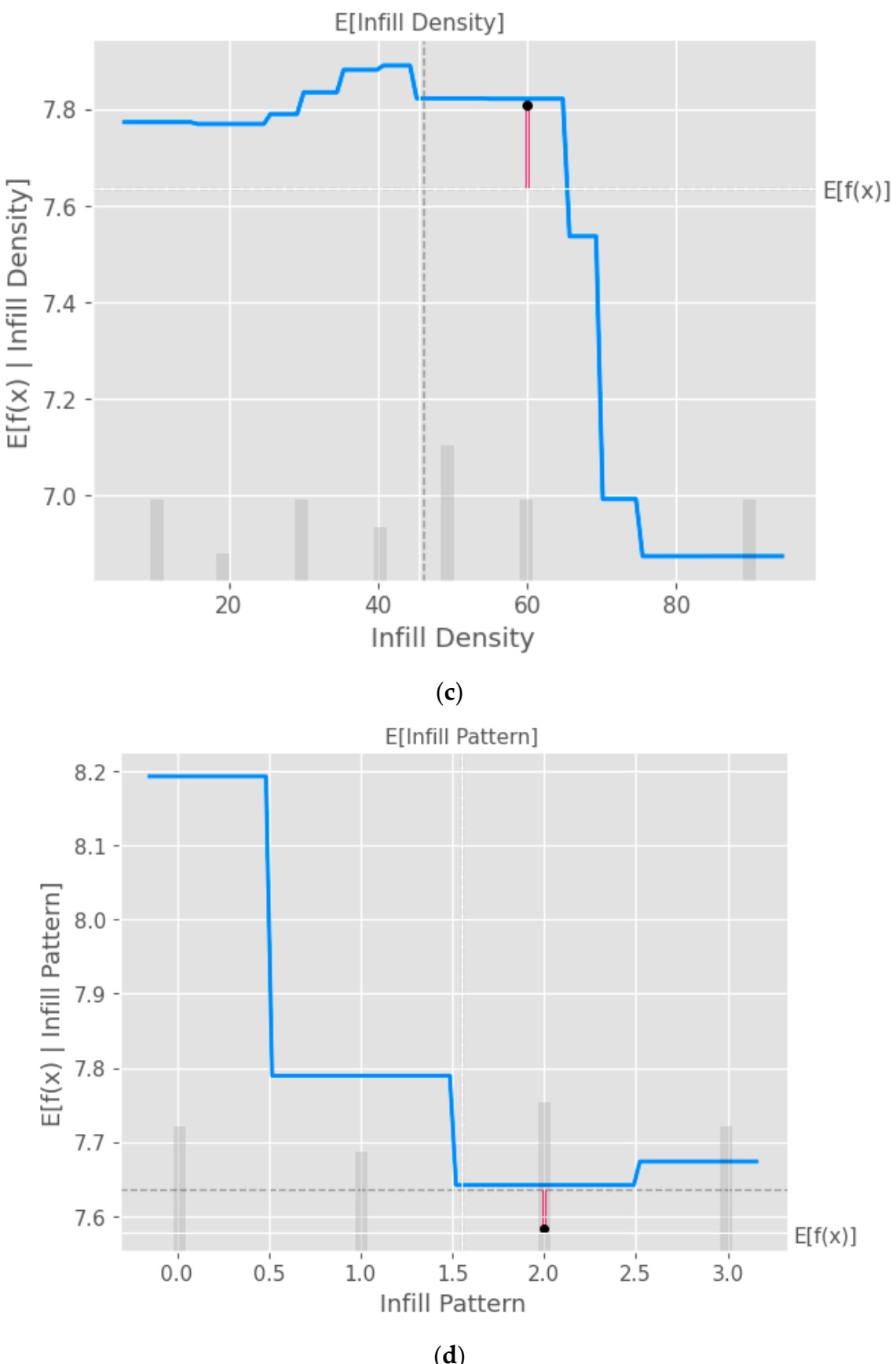

(**c**)

(**d**)

**Figure 10.** *Cont.*

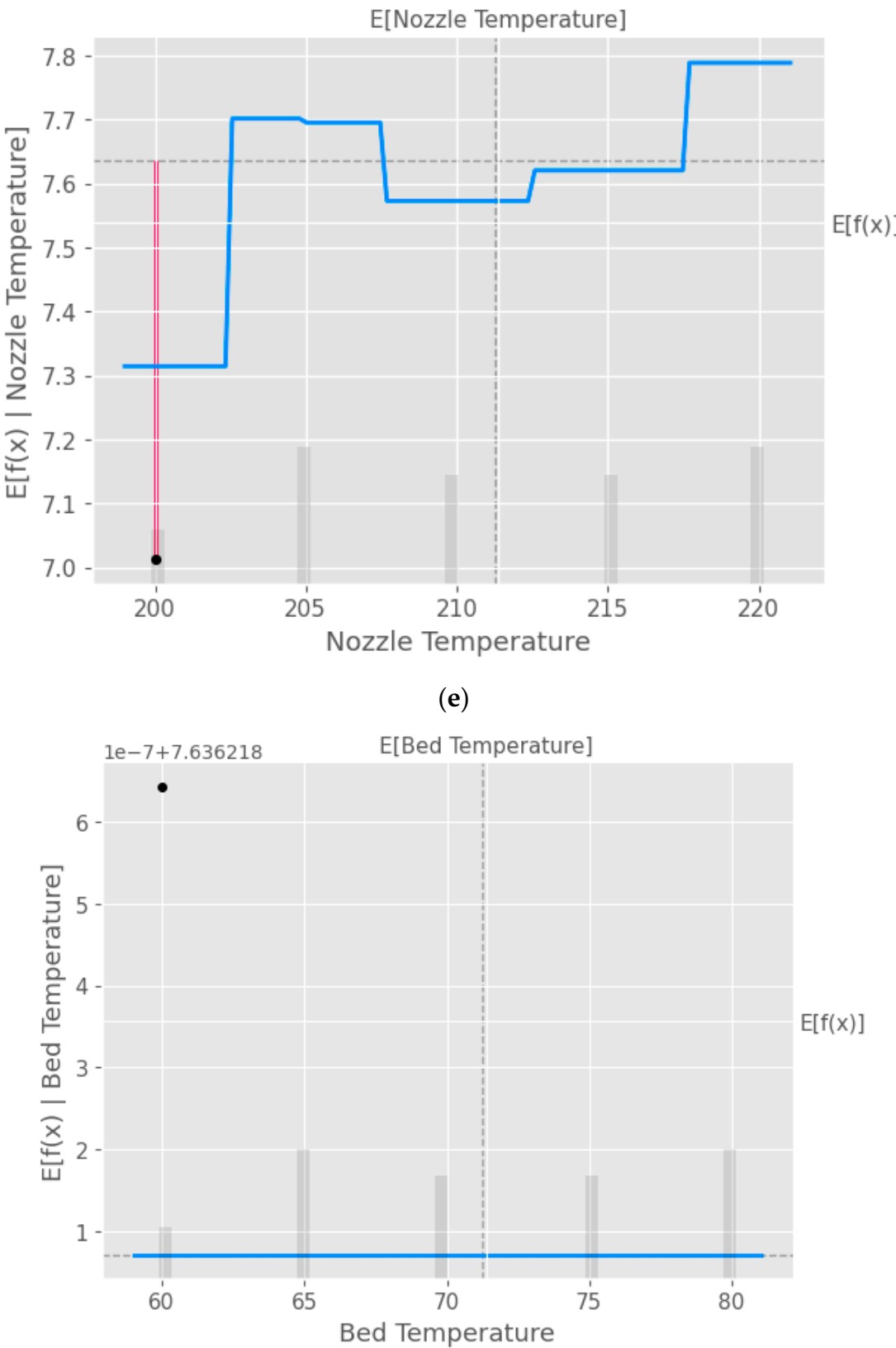

(**e**)

(**f**)

**Figure 10.** *Cont.*

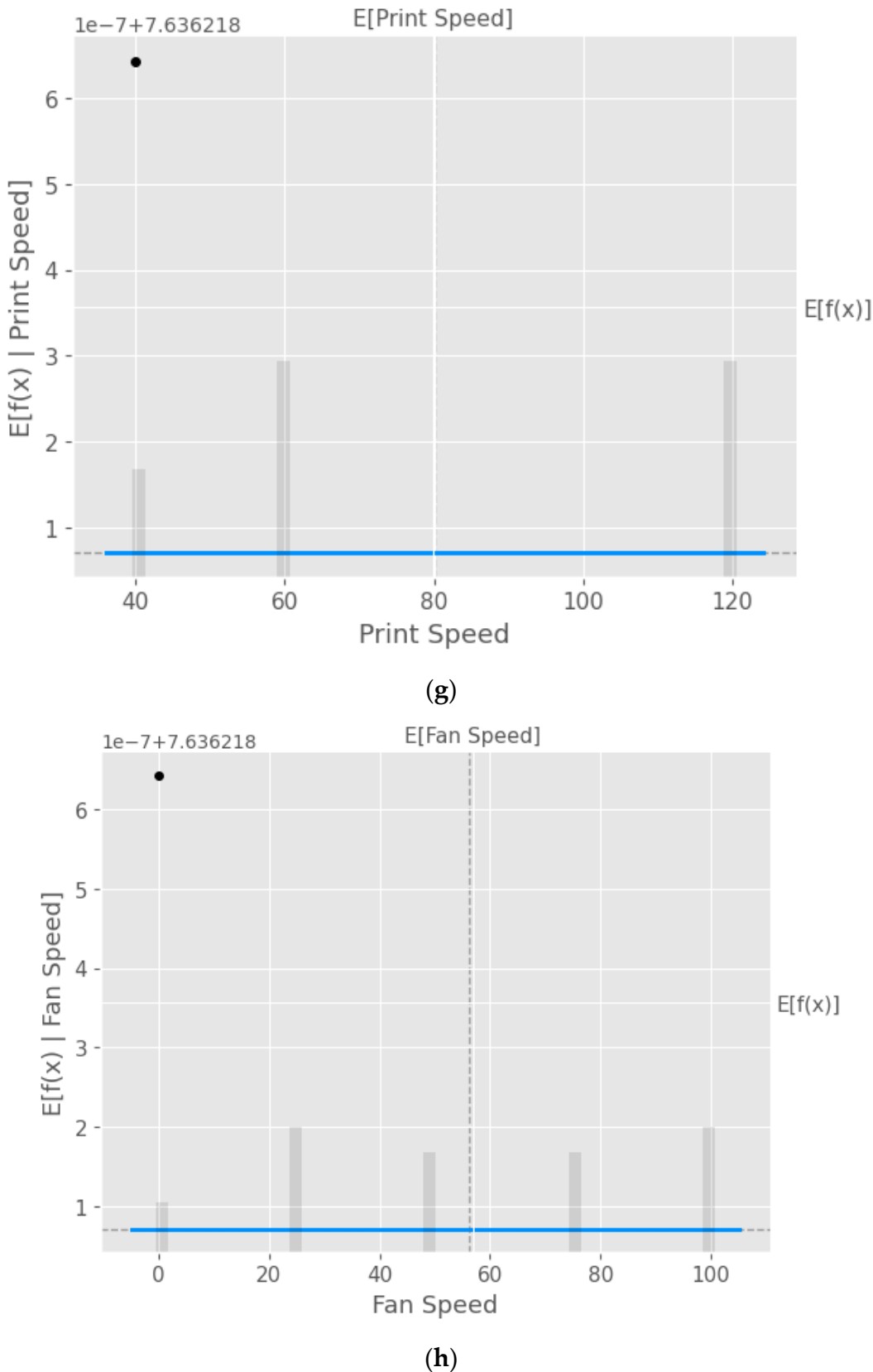

**Figure 10.** Partial dependence plots (PDP) with XGB model taking into consideration the input parameters (**a**) layer height, (**b**) wall thickness, (**c**) infill density, (**d**) infill pattern, (**e**) nozzle temperature, (**f**) bed temperature, (**g**) print speed, and (**h**) fan speed.

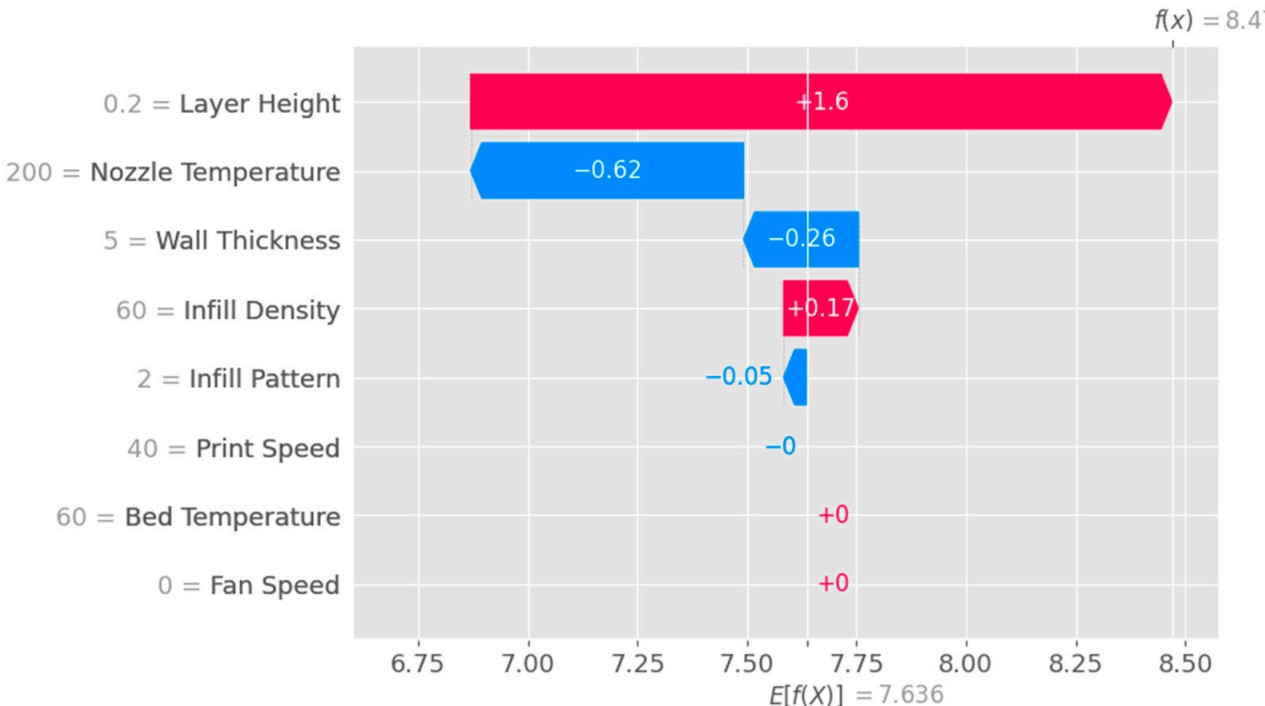

**Figure 11.** Waterfall plot obtained with XGB model.

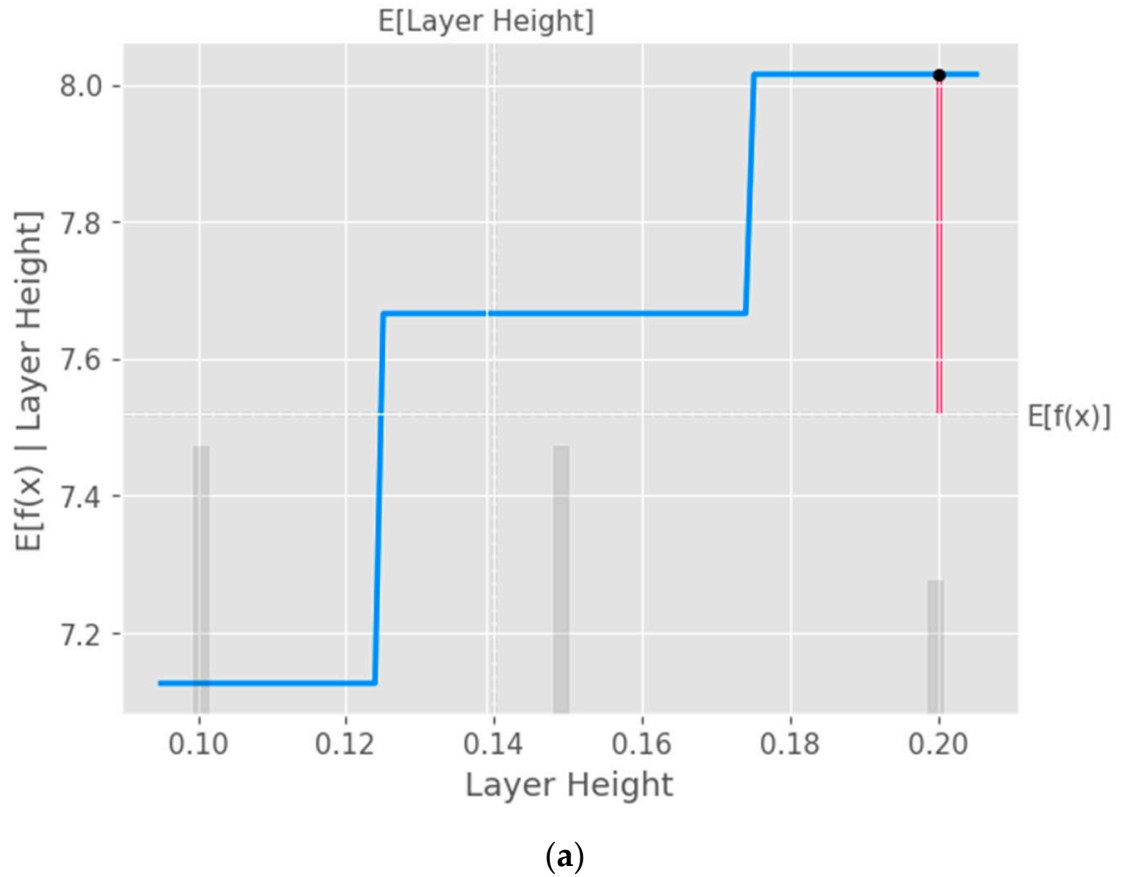

(**a**)

**Figure 12.** *Cont.*

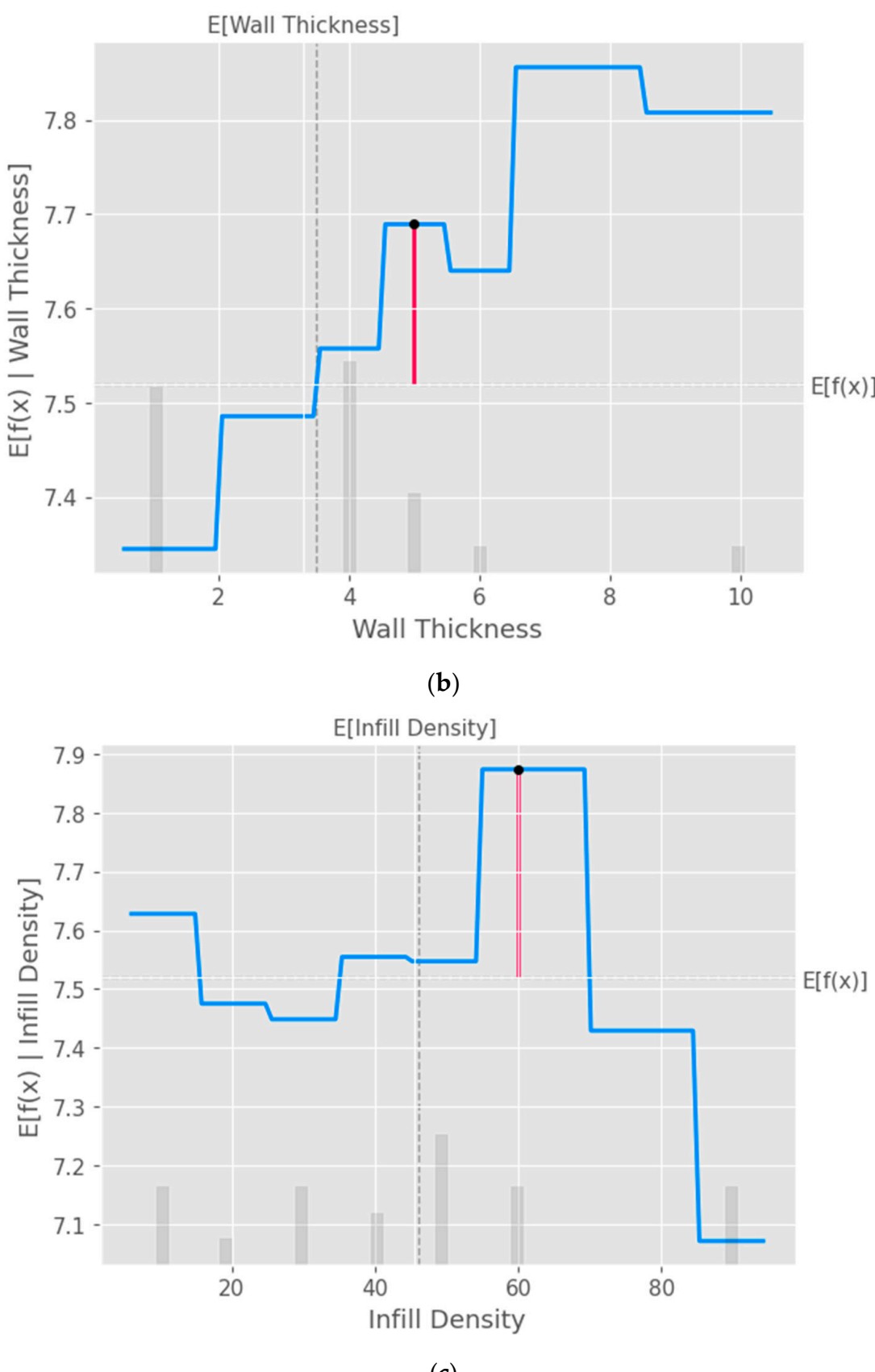

(**b**)

(**c**)

**Figure 12.** *Cont.*

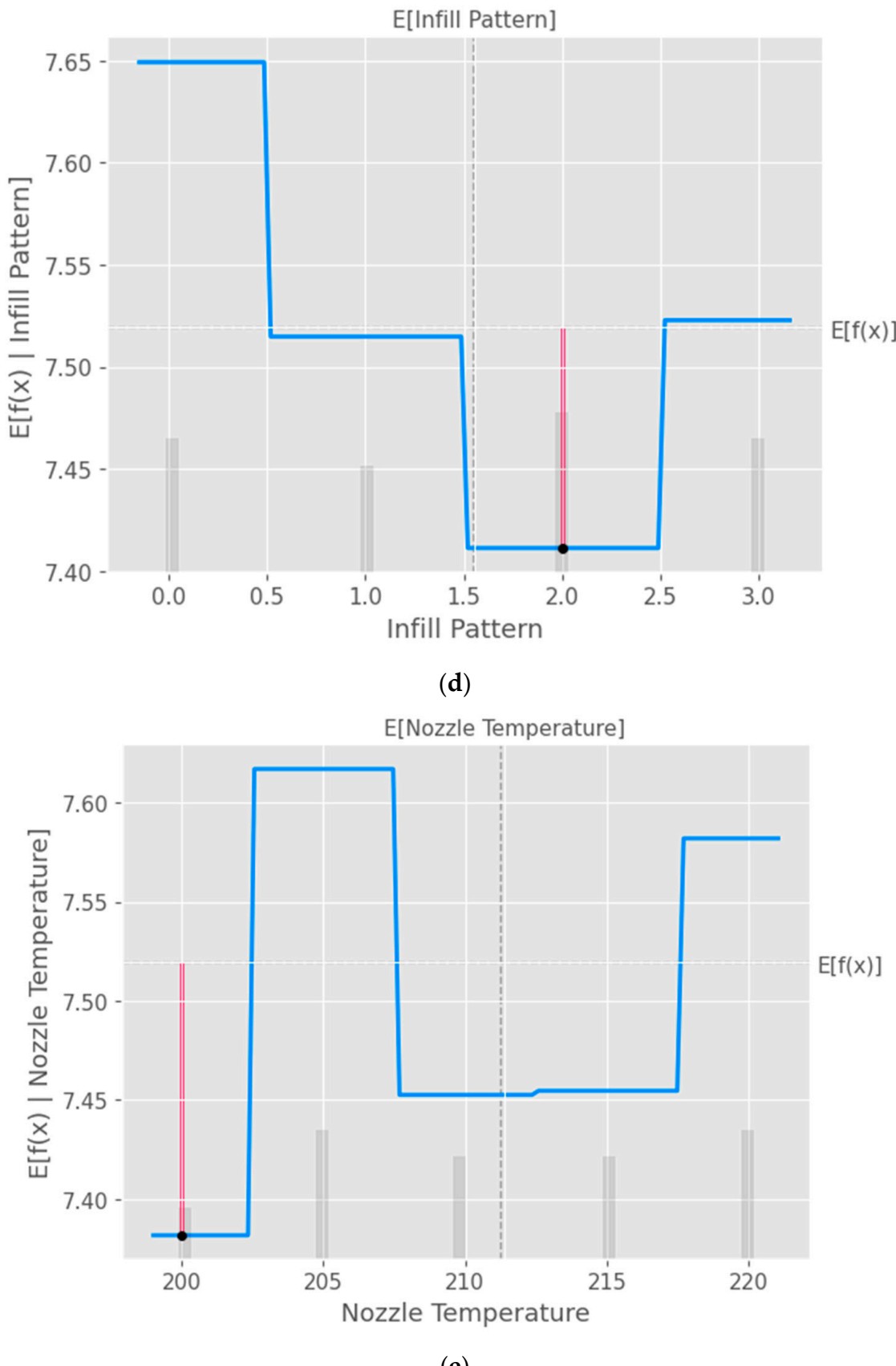

(**d**)

(**e**)

**Figure 12.** *Cont.*

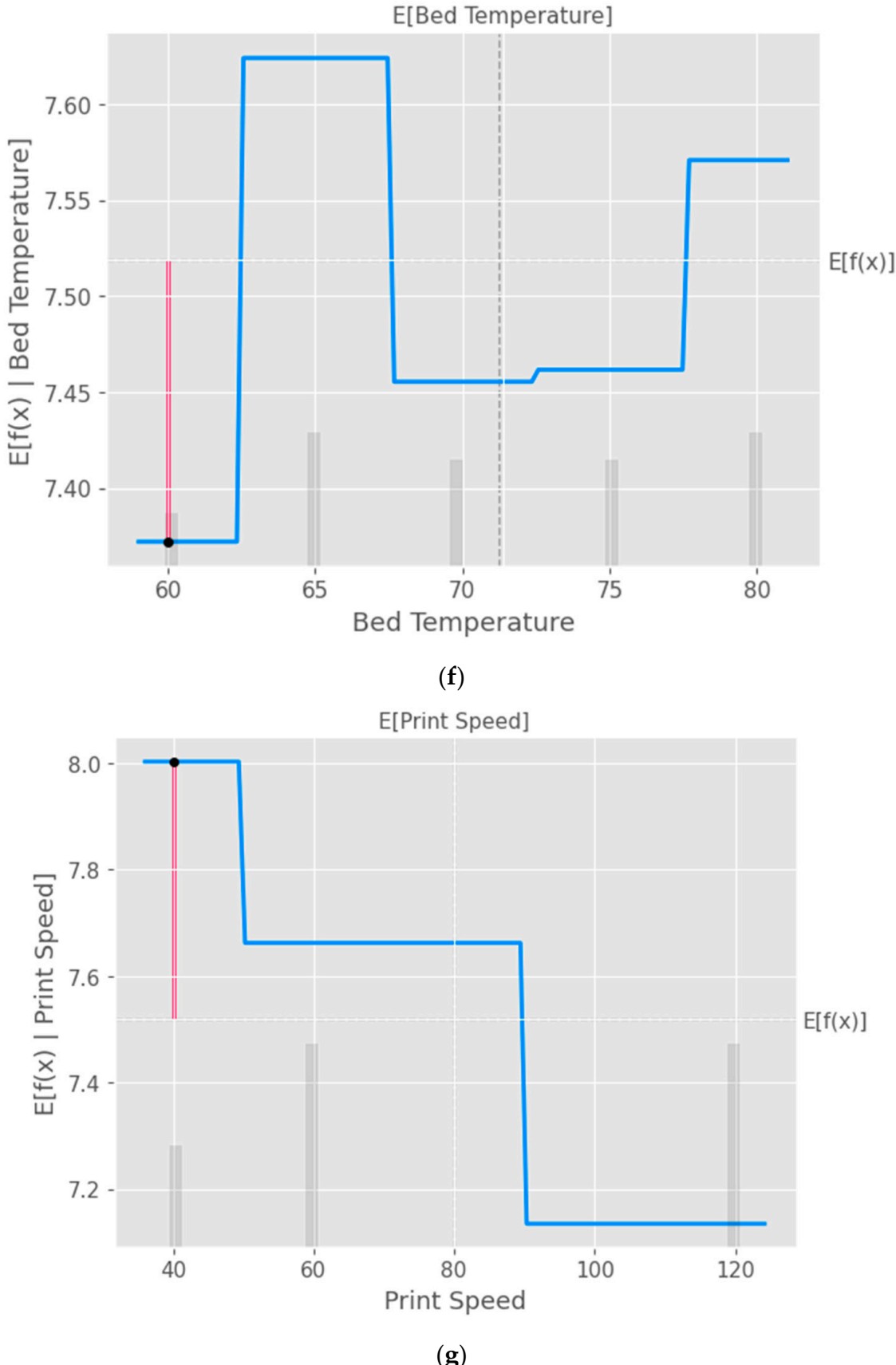

(**f**)

(**g**)

**Figure 12.** *Cont.*

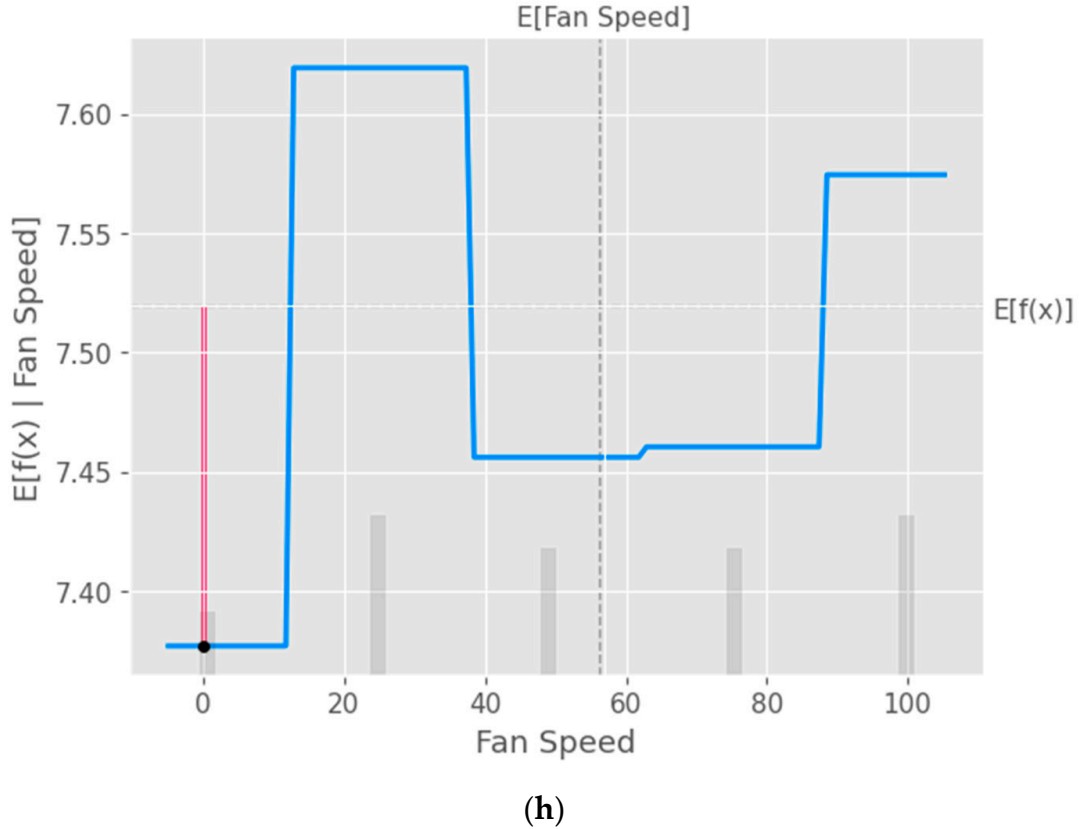

(**h**)

**Figure 12.** Partial dependence plots (PDPs) with EBM taking into consideration the input parameters (**a**) layer height, (**b**) wall thickness, (**c**) infill density, (**d**) infill pattern, (**e**) nozzle temperature, (**f**) bed temperature, (**g**) print speed, and (**h**) fan speed.

Figure 13 shows a waterfall plot using SHAP values obtained from the Explainable Boosting Machine (EBM) model.

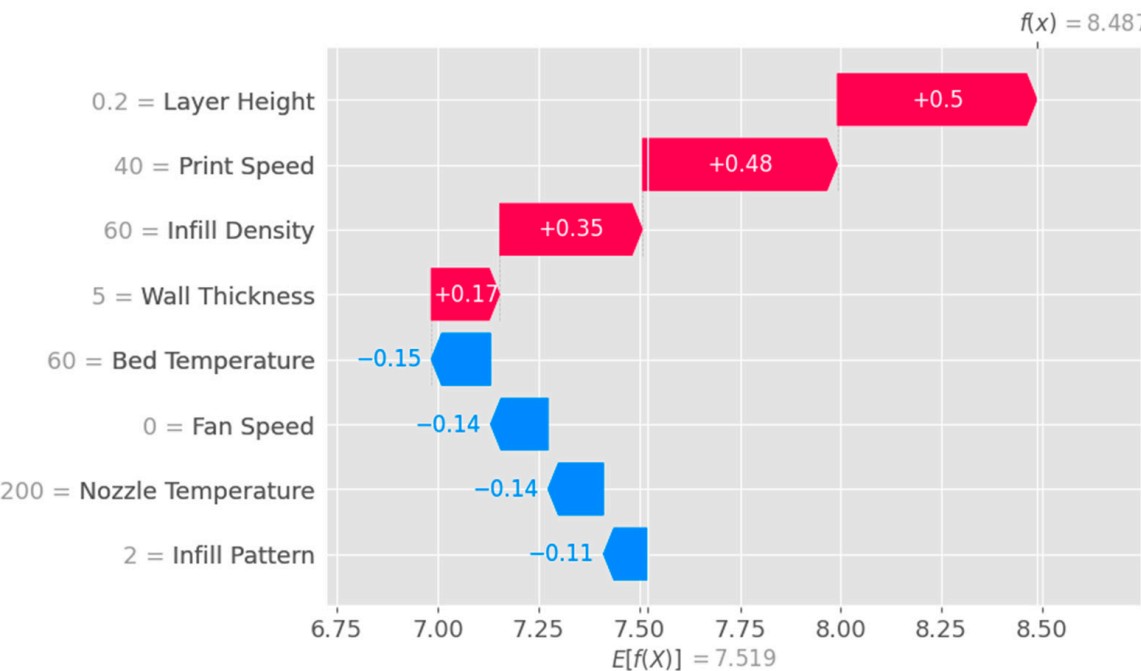

**Figure 13.** Waterfall plot with EBM Model.

The main difference between the two waterfall plots lies in the model they are based on. The first waterfall plot is created using the SHAP values obtained from the XGBoost model, while the second waterfall plot is created using the SHAP values obtained from the Explainable Boosting Machine (EBM) model.

XGBoost and the EBM are both powerful ensemble learning techniques that provide accurate predictions. While XGBoost builds complex models by iteratively fitting weak learners, typically Decision Trees, the EBM combines simple models additively to provide more interpretable results. The EBM is specifically designed to be more interpretable than other boosting methods, including XGBoost. While XGBoost can provide feature importance scores, the EBM offers a more granular understanding of each feature's effect on the target variable. XGBoost is known for its speed, scalability, and high predictive performance, whereas EBM provides a good trade-off between accuracy and interpretability. The model complexity and interpretability of XGBoost and EBM are the primary differences between them. The choice between them depends on the specific requirements of the task at hand, and both models have their strengths and weaknesses.

The beeswarm plot obtained from the XGB model, as shown in Figure 14, is a powerful visualization tool that displays the SHAP values for all features and instances in the dataset. It helps to understand the overall impact of each feature on the model's predictions and offers insights into the distribution of feature contributions across all instances.

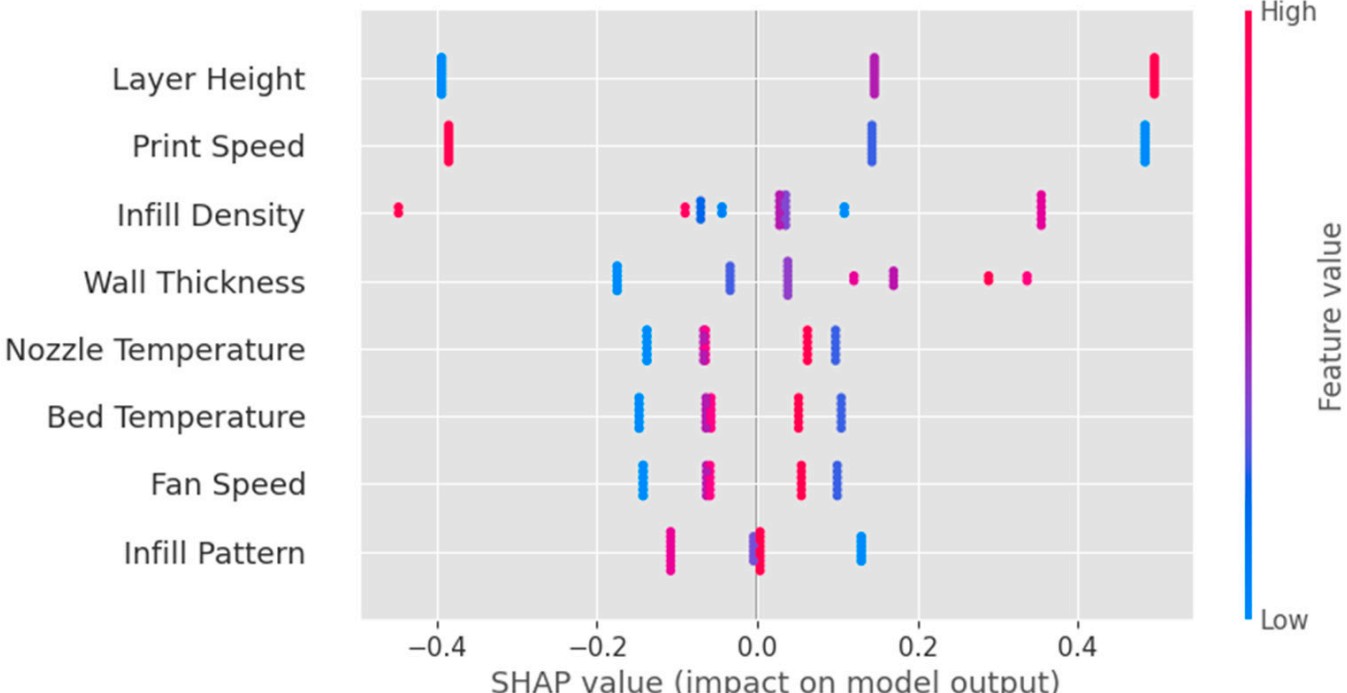

**Figure 14.** Beeswarm plot obtained from XGB model.

The heatmap plot obtained from the XGB model, as shown in Figure 15, is a visualization tool that displays SHAP values for all features and a subset of instances in the dataset as a heatmap. It helps to understand the impact of each feature on the model's predictions and offers insights into the distribution of feature contributions across the selected instances.

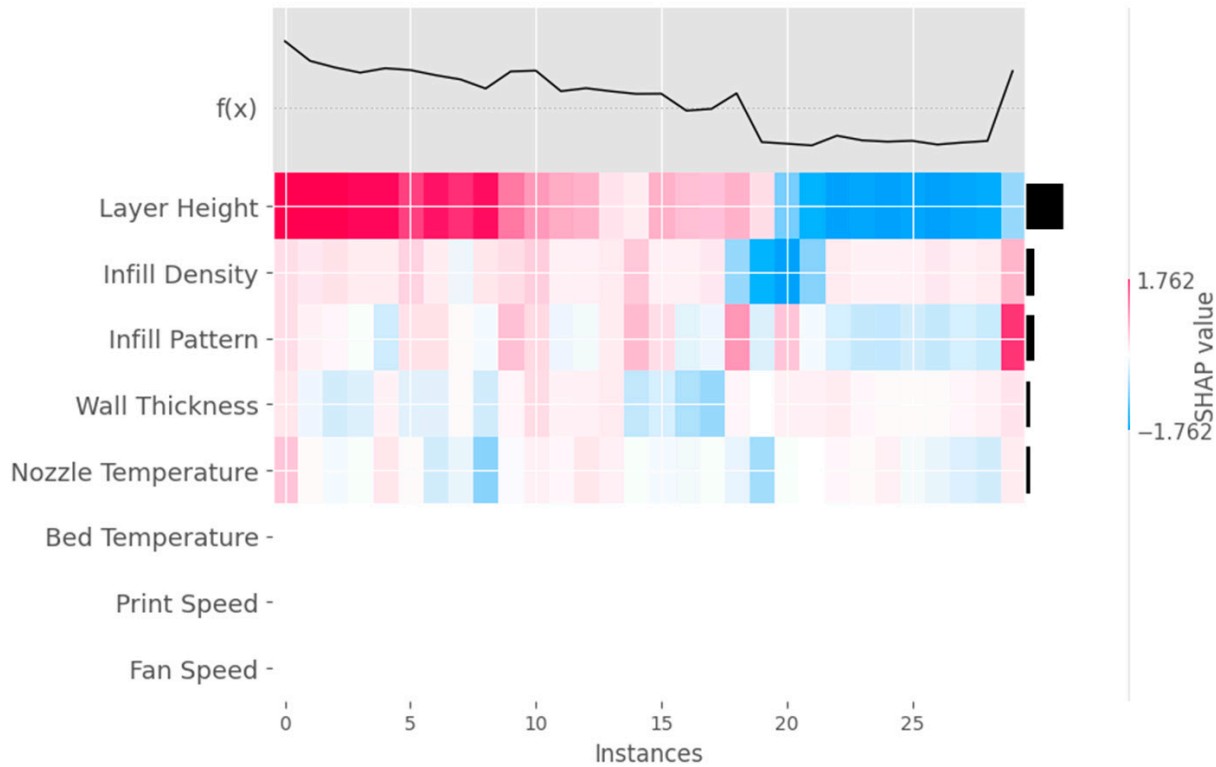

**Figure 15.** Heatmap plot obtained from XGB model.

## 4. Discussion

From the obtained results, it is observed that the XGBoost algorithm outperforms the other machine learning algorithms by resulting in the highest $R^2$ value. XGBoost is a highly effective machine learning algorithm due to its scalability, regularization techniques, speed, handling of missing data, feature importance, and flexibility. It is optimized for parallel processing on multi-core CPUs, allowing it to handle large datasets with millions of examples and features efficiently. XGBoost uses both L1 and L2 regularization techniques to prevent overfitting and improve the model's generalization performance, and it can handle missing data effectively using gradient-based sampling. XGBoost provides a measure of feature importance, allowing for the identification of significant features in the dataset, and it is flexible, handling both regression and classification tasks. These features make XGBoost a popular choice for many machine learning tasks, and it is often used as a benchmark for other algorithms.

Explainable AI (XAI) is of paramount importance in the present research work for several reasons. In the context of predicting surface roughness for additively manufactured specimens, employing XAI techniques can enhance the understanding, trust, and effectiveness of the developed models, which, in turn, can lead to better decision making and improved outcomes of the model's predictions. This interpretability enables researchers to gain insights into the relationships between input features and the target variable (surface roughness), leading to a better understanding of the underlying physical processes involved in additive manufacturing. By providing clear explanations of how a model makes predictions, XAI enhances the trust that stakeholders have in the model. This is crucial in engineering applications, such as additive manufacturing, where the quality and performance of produced components are vital. Trustworthy models can facilitate the adoption of AI-driven solutions in the industry and help ensure that the developed models are used effectively. XAI techniques can aid in validating the developed models by revealing the contributions of each feature and detecting any biases or inconsistencies. By examining the importance of the features and potential interaction effects, researchers can evaluate the models and identify any areas that require further improvement, ul-

timately leading to more accurate and reliable predictions. XAI enables researchers to communicate their findings more effectively, both within the research community and to industry stakeholders. Clear, interpretable visualizations, such as partial dependence plots, beeswarm plots, and heatmap plots, can convey complex relationships and model behaviors in an accessible manner. This facilitates better collaboration and understanding among researchers, engineers, and decision-makers involved in the additive manufacturing process. In some industries, including additive manufacturing, regulatory compliance may require explanations for AI-driven decisions. XAI techniques can provide the necessary transparency and interpretability to meet these regulatory requirements, ensuring that AI solutions can be successfully implemented in real-world applications.

### 5. Conclusions

In conclusion, this research work presented a comprehensive analysis of the prediction of surface roughness in additively manufactured polylactic acid (PLA) specimens using eight different supervised machine learning regression-based algorithms. The results demonstrate the superiority of the XGBoost algorithm, with the highest coefficient of determination value of 0.9634, indicating its ability to accurately predict surface roughness. Additionally, this study pioneers the use of explainable AI techniques to enhance the interpretability of machine learning models, offering valuable insights into feature importance, interaction effects, and model behavior. The comparative analysis of the algorithms, combined with the explanations provided via explainable AI, contributes to a better understanding of the relationship between surface roughness and structural integrity in additive manufacturing.

**Author Contributions:** Conceptualization, A.M., V.S.J., E.M.S. and S.P.; methodology, A.M.; software, V.S.J.; validation, S.P., E.M.S. and A.M.; formal analysis, S.P.; investigation, V.S.J.; resources, S.P.; data curation, V.S.J.; writing—original draft preparation, A.M.; writing—review and editing, E.M.S.; visualization, S.P.; supervision, V.S.J.; project administration, V.S.J. All authors have read and agreed to the published version of the manuscript.

**Funding:** This research received no external funding.

**Data Availability Statement:** Data are available upon request by readers.

**Conflicts of Interest:** The authors declare no conflict of interest.

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
