# Peer review of "Explainable Artificial Intelligence (XAI) and Supervised Machine Learning-based Algorithms for Prediction of Surface Roughness of Additively Manufactured Polylactic Acid (PLA) Specimens"

_2673-3161, doi:10.3390/applmech4020034_

Round 1

Reviewer 1 Report

Dear Authors,

The manuscript covers quite an interesting topic, linked to the need in 3D printing to have a prediction of important factors such as surface roughness. In fact, the authors have chosen to focus on the prediction of surface roughness of 3D printed PLA samples using several supervised machine learning regression-based algorithms. However, the research set forth in this way in the manuscript appears to be lacking to be considered for publication at its current form.

Some comments of the reviewer that may be useful to the authors are reported.

Line 18: as can be seen from the manuscript, this part is out of scope of the work. In fact, no correlations between surface roughness and structural integrity are performed. Therefore, consider deleting this part and inserting it, as is currently done, in the conclusions.

Lines 59-67: rewrite this part, because there are redundant sentences

Line 97: Explain why the printing parameters shown in Table 1 were chosen (consider including the bibliography, currently too short).

Furthermore, the need to reduce time and materials is understood, but why were not at least 3 replications considered? As is known, the 3D printing process (especially desktop-type FFF) is highly unreliable and not very reproducible, therefore considering only 1 sample as representative of a certain set of parameters seems to be an unrobust assumption.

Line 102: Is the material used the same throughout the research work? From Figure 1 it is possible to see 3 different colors: specify the reason. The same material should be used (including additives/colourants). However, if these have an insignificant influence on the surface roughness, it must be explained, quoting the appropriate sources in the literature.

Row 105: how were the surface roughness measurements taken?

Line 174 (“confusion”): check

Line 207: Table 2, missing

Line 489: Check

Kind Regards

Author Response

Dear Reviewer,

We are grateful for your critical comments in order to improve the quality of the paper. We have carried out the required correction in the revised file and we have addressed your queries below:

  1. Line 18: as can be seen from the manuscript, this part is out of scope of the work. In fact, no correlations between surface roughness and structural integrity are performed. Therefore, consider deleting this part and inserting it, as is currently done, in the conclusions.

Comment: This part has been deleted.

  1. Lines 59-67: rewrite this part, because there are redundant sentences

Comment: The part has been re-written.

  1. Line 97: Explain why the printing parameters shown in Table 1 were chosen (consider including the bibliography, currently too short).

Comment:  We have extended the bibliography part by referencing the papers [25-28] used for determining the printing parameters.

  1. Furthermore, the need to reduce time and materials is understood, but why were not at least 3 replications considered? As is known, the 3D printing process (especially desktop-type FFF) is highly unreliable and not very reproducible, therefore considering only 1 sample as representative of a certain set of parameters seems to be an unrobust assumption.

Comment: Using Mitutoyo SJ-10 surface roughness tester measurements were taken at four locations and their average was considered.

  1. Line 102: Is the material used the same throughout the research work? From Figure 1 it is possible to see 3 different colors: specify the reason. The same material should be used (including additives/colourants). However, if these have an insignificant influence on the surface roughness, it must be explained, quoting the appropriate sources in the literature.

Comment: The other factors such as colour material and colour percent were neglected as these fac tors have negligible effect on surface roughness [40].

  1. Row 105: how were the surface roughness measurements taken?

Comment: Using Mitutoyo SJ-10 surface roughness tester measurements were taken at four locations and their average was considered.

  1. Line 174 (“confusion”): check

Comment: The correction has been carried out.

  1. Line 207: Table 2, missing

Comment: Table 2 has been included.

  1. Line 489: Check

Comment: Correction has been carried out.

Reviewer 2 Report

SUMMARY

The paper explores a machine learning-based approach to predict the surface roughness of 3D-printed Polyactic Acid (PLA) specimens. Eight supervised machine-learning regression-based algorithms are tested for their predictability of surface roughness from eight process parameters. The dataset used for training and validating the models is obtained from an RSM DOE to generate 30 different trials of 3D-printed standard dog-bone specimens. It is concluded that from the eight different MA algorithms employed in the study, the XG boost algorithm yield the best predictions with the highest R2 value. Explainable AI (XAI) visualization methods using Shapley values are also presented to highlight the decision-making process and the influence of each process parameter on the surface roughness prediction.

CRITICAL EVALUATION AND COMMENTS

The reviewer appreciates the authors’ efforts at presenting a detailed analysis of the use of both supervised ML models and XAI in predicting the surface roughness of AM specimens, especially the significance of XAI in explaining the model predictions and the role of individual process parameters. However, there are several concerns as described in the comments below.

General concept comments

1. The reviewer was able to find several articles among available literature with similar work on ML prediction of surface roughness in AM specimens. However, no such literature was found to be cited in this manuscript. The authors are strongly recommended to cite such relevant papers to enhance the description of the novelty of their work. Please find some of the recommended citations below:

a. Prediction of surface roughness in extrusion-based additive manufacturing with machine learning (Li et al., DOI: https://doi.org/10.1016/j.rcim.2019.01.004).

b. Surface roughness prediction in additive manufacturing using machine learning (Wu et al., DOI: https://doi.org/10.1115/MSEC2018-6501)

c. Prediction of metal additively manufactured surface roughness using deep neural network (So et al., DOI: https://doi.org/10.3390/s22207955).

d. Predicting and optimizing the surface roughness of additive manufactured parts using an artificial neural network model and genetic algorithm (Ulkir and Akgun, DOI: https://doi.org/10.1080/13621718.2023.2200572).  

2. In section 2 (materials and methods), the authors are requested to describe the experimental method of measuring the surface roughness of their specimens.

3. Section 3.1 (supervised machine learning algorithms) needs major revision. Mathematical descriptions of the eight algorithms in separate subsections and supporting citations are strongly recommended. Figures 6 and 7 may be a part of the decision tree regression sub section.

4. Table 2 is missing from the manuscript. Please add it in the revised version.

5. In section 3.2 (explainable artificial intelligence (XAI) approach), the authors are requested to cite the appropriate paper for the use of SHAP (https://dl.acm.org/doi/10.5555/3295222.3295230).

6. Authors are requested to provide a clarification on why a simple linear regression and EBM model are chosen to explain the prediction for a specific instance using SHAP in section 3.2, considering that none of the two models have been evaluated in section 3.1. This introduces a disconnect between sections 3.1 and 3.2. In the reviewer’s opinion, explaining an XG boost prediction for an instance would make better sense, since it has been identified as the best algorithm in section 3.1.

Specific line comments

1. Lines 142-157 provide a verbose description of MAE, MSE and R2 without adding any real value to the paper. Please consider just providing a mathematical description (formulae) and omitting the textual content.

2. In line 174, the correlation matrix heatmap is erroneously referred to as confusion heat map. The authors are requested to correct this.

3. In lines 187-190, it is explained that positive correlation is represented by a shade of blue, while negative correlation is represented by a shade of red. However, in Figure 5, the shading is exactly opposite of the description. The authors are requested to correct this error.

The quality of English is acceptable, with few minor revisions needed. Specifically, please use past tense in line 117, and change "Pandas library used" to "Pandas library was used" in line 120.

Author Response

Dear Reviewer,

We are grateful for your critical comments in order to improve the quality of the paper. We have carried out the required correction in the revised file and we have addressed your queries below:

  1. The reviewer was able to find several articles among available literature with similar work on ML prediction of surface roughness in AM specimens. However, no such literature was found to be cited in this manuscript. The authors are strongly recommended to cite such relevant papers to enhance the description of the novelty of their work. Please find some of the recommended citations below:
  2. Prediction of surface roughness in extrusion-based additive manufacturing with machine learning (Li et al., DOI: https://doi.org/10.1016/j.rcim.2019.01.004).
  3. Surface roughness prediction in additive manufacturing using machine learning (Wu et al., DOI: https://doi.org/10.1115/MSEC2018-6501)
  4. Prediction of metal additively manufactured surface roughness using deep neural network (So et al., DOI: https://doi.org/10.3390/s22207955).
  5. Predicting and optimizing the surface roughness of additive manufactured parts using an artificial neural network model and genetic algorithm (Ulkir and Akgun, DOI: https://doi.org/10.1080/13621718.2023.2200572).  

Comment: Thanks for suggesting the given recommendations. We have included these recommendations as citations in our revised file.

  1. In section 2 (materials and methods), the authors are requested to describe the experimental method of measuring the surface roughness of their specimens.

Comment: Using Mitutoyo SJ-10 surface roughness tester measurements were taken at four locations and their average was considered.

  1. Section 3.1 (supervised machine learning algorithms) needs major revision. Mathematical descriptions of the eight algorithms in separate subsections and supporting citations are strongly recommended. Figures 6 and 7 may be a part of the decision tree regression sub section.

Comment: The explanation of each ML algorithms is included in sub-section wise with mathematical formulations and also the mentioned plot i.e., Figure 6 and 7 is included in Decision tree sub section.

  1. Table 2 is missing from the manuscript. Please add it in the revised version.

Comment: Table 2 is included in the revised file.

  1. In section 3.2 (explainable artificial intelligence (XAI) approach), the authors are requested to cite the appropriate paper for the use of SHAP (https://dl.acm.org/doi/10.5555/3295222.3295230).

Comment: Included

  1. Authors are requested to provide a clarification on why a simple linear regression and EBM model are chosen to explain the prediction for a specific instance using SHAP in section 3.2, considering that none of the two models have been evaluated in section 3.1. This introduces a disconnect between sections 3.1 and 3.2. In the reviewer’s opinion, explaining an XG boost prediction for an instance would make better sense, since it has been identified as the best algorithm in section 3.1.

Comment: This was a vey good point recommended by the reviewer. We have included the mse, mae and r-square value results for EBM in our revised model and linear model is replaced with XGboost model for comparing it with EBM model.

Specific line comments

  1. Lines 142-157 provide a verbose description of MAE, MSE and R2without adding any real value to the paper. Please consider just providing a mathematical description (formulae) and omitting the textual content.

Comment: The formulas are now included in the revised part.

  1. In line 174, the correlation matrix heatmap is erroneously referred to as confusion heat map. The authors are requested to correct this.

Comment: Correction has been carried out.

  1. In lines 187-190, it is explained that positive correlation is represented by a shade of blue, while negative correlation is represented by a shade of red. However, in Figure 5, the shading is exactly opposite of the description. The authors are requested to correct this error.

Comment: Correction has been carried out

Round 2

Reviewer 2 Report

I thank the authors for promptly addressing the comments made in the first review report. After reviewing the revised version, only the following minor concerns remain to be addressed:

1. The authors are still strongly recommended to omit the textual description of MAE, MSE and R^2 in lines 148-163 from the paper. It is the reviewer's opinion that the description of these errors does not add value to the paper since the pertinent audience should already be knowledgeable enough to understand their meaning and significance.

2. While the authors added the mathematical description of the errors in the paper, they were added as part of the wrong sub-section (sec 3.1.8). The authors are requested to move lines 364-373 after the completion of all sub-sections in sec 3.1, i.e., after sec 3.1.9.

3. The authors are requested to further emphasize that the PDPs in Figure 9 and waterfall plot in Figure 10 were created from Shapley values obtained from XG boost model. This may be done in line 421 and in captions of Figs. 9 and 10. The authors have mentioned EBM model in Figs. 11 and 12 but not the XG boost model in Figs. 9 and 10. Although the difference between the waterfall plots (Figs. 10 and 12) in terms of model use is mentioned in lines 498-501, it seems a bit too late and the reader should be made aware of this difference earlier in the section. 

4. Please mention the model used for creating Figures 13 and 14.

Please correct the spelling of correlation in line 172. 

Author Response

Dear Reviewer,

We express our gratitude for your insightful comments and the time you've dedicated to reviewing our manuscript. Your suggestions have significantly contributed to improving the quality of our work.

In response, we have meticulously reviewed each point you raised and have addressed them in detail below.

  1. The authors are still strongly recommended to omit the textual description of MAE, MSE and R^2 in lines 148-163 from the paper. It is the reviewer's opinion that the description of these errors does not add value to the paper since the pertinent audience should already be knowledgeable enough to understand their meaning and significance.

Comment: In the revised file we have omitted the textual description of these metric features.

  1. While the authors added the mathematical description of the errors in the paper, they were added as part of the wrong sub-section (sec 3.1.8). The authors are requested to move lines 364-373 after the completion of all sub-sections in sec 3.1, i.e., after sec 3.1.9.

Comment: The lines dealing with the mathematical description is moved after the completion of all sub-sections.

  1. The authors are requested to further emphasize that the PDPs in Figure 9 and waterfall plot in Figure 10 were created from Shapley values obtained from XG boost model. This may be done in line 421 and in captions of Figs. 9 and 10. The authors have mentioned EBM model in Figs. 11 and 12 but not the XG boost model in Figs. 9 and 10. Although the difference between the waterfall plots (Figs. 10 and 12) in terms of model use is mentioned in lines 498-501, it seems a bit too late and the reader should be made aware of this difference earlier in the section.

Comment: We have modified our captions highlighting which model was used i.e., XGB model.

  1. Please mention the model used for creating Figures 13 and 14.

Comment: XGB Model was used. We have included it in the caption.

  1. Please correct the spelling of the correlation in line 172.

Comment: Spelling has been corrected.
